# Identification of a Potent Cytotoxic Pyrazole with Anti-Breast Cancer Activity That Alters Multiple Pathways

**DOI:** 10.3390/cells11020254

**Published:** 2022-01-12

**Authors:** Denisse A. Gutierrez, Lisett Contreras, Paulina J. Villanueva, Edgar A. Borrego, Karla Morán-Santibañez, Jessica D. Hess, Rebecca DeJesus, Manuel Larragoity, Ana P. Betancourt, Jonathon E. Mohl, Elisa Robles-Escajeda, Khodeza Begum, Sourav Roy, Robert A. Kirken, Armando Varela-Ramirez, Renato J. Aguilera

**Affiliations:** 1Cellular Characterization and Biorepository Core Facility, Border Biomedical Research Center, Department of Biological Sciences, College of Science, The University of Texas at El Paso, 500 West University Avenue, El Paso, TX 79968-0519, USA; dagutierrez5@utep.edu (D.A.G.); lcontreras4@miners.utep.edu (L.C.); pjvillanueva@miners.utep.edu (P.J.V.); eaborregopu@miners.utep.edu (E.A.B.); karla.moransn@uanl.edu.mx (K.M.-S.); jdhess@miners.utep.edu (J.D.H.); repellerano@miners.utep.edu (R.D.); mlarragoity@miners.utep.edu (M.L.); abetancourt@utep.edu (A.P.B.); erobles3@utep.edu (E.R.-E.); kbegum@utep.edu (K.B.); sroy1@utep.edu (S.R.); rkirken@utep.edu (R.A.K.); avarela2@utep.edu (A.V.-R.); 2Department of Bioinformatics, The University of Texas at El Paso, 500 West University Avenue, El Paso, TX 79968-0519, USA; jemohl@utep.edu

**Keywords:** pyrazole, triple-negative breast cancer, cytotoxicity, apoptosis, ROS, kinase activity regulation, microtubule disruption, RNA-seq

## Abstract

In this study, we identified a novel pyrazole-based derivative (P3C) that displayed potent cytotoxicity against 27 human cancer cell lines derived from different tissue origins with 50% cytotoxic concentrations (CC_50_) in the low micromolar and nanomolar range, particularly in two triple-negative breast cancer (TNBC) cell lines (from 0.25 to 0.49 µM). In vitro assays revealed that P3C induces reactive oxygen species (ROS) accumulation leading to mitochondrial depolarization and caspase-3/7 and -8 activation, suggesting the participation of both the intrinsic and extrinsic apoptotic pathways. P3C caused microtubule disruption, phosphatidylserine externalization, PARP cleavage, DNA fragmentation, and cell cycle arrest on TNBC cells. In addition, P3C triggered dephosphorylation of CREB, p38, ERK, STAT3, and Fyn, and hyperphosphorylation of JNK and NF-kB in TNBC cells, indicating the inactivation of both p38MAPK/STAT3 and ERK1/2/CREB signaling pathways. In support of our in vitro assays, transcriptome analyses of two distinct TNBC cell lines (MDA-MB-231 and MDA-MB-468 cells) treated with P3C revealed 28 genes similarly affected by the treatment implicated in apoptosis, oxidative stress, protein kinase modulation, and microtubule stability.

## 1. Introduction

At present, cancer causes one in every six deaths worldwide, being the second leading cause of death globally [1]. By 2040, new global cancer cases are expected to increase from 17 to 27.5 million (an increase of 161%) [2]. The need for new pharmacological agents, possibly involving new therapeutic mechanisms of action to overcome drug resistance and minimize off-target secondary effects, is of high importance.

Pyrazoles, nitrogen-containing heterocyclic compounds, have widespread pharmacological activities [3]. They have been recognized to display anti-inflammatory [4], antimicrobial [5], antiviral [6], antimalarial [7], antipsychotic [8], antiglaucoma [9], antihypertensive [10], anti-Alzheimer [11], anti-Parkinson [12], and other activities. Therefore, the pyrazole moiety is considered indispensable in the structural design of new pharmacologically active agents [3,13].

Many pyrazole-based analogs with promising anticancer activity have been identified [14]. These have been developed through molecular modeling and docking studies that seek to inhibit specific targets important for cancer management such as tumor growth factors and protein kinases [3,13]. Moreover, several groups have reported that pyrazole derivatives exert antiproliferative and cytotoxic activities on several cancer cell lines with promising clinical activity [15,16,17].

In the present study, a pyrazole derivative, N′-[(2-hydroxy-1-naphthyl) methylene]-1,4,5,6-tetrahydrocyclopenta[c]pyrazole-3-carbohydrazide (P3C), is described with potent anticancer activities. P3C was initially identified as a cytotoxic pyrazole from a high-throughput screen (HTS) comprising 4640 chemical compounds from the ChemBridge DIVERset library tested on the human TNBC MDA-MB-231 cell line. After its initial detection, P3C cytotoxicity was confirmed in 27 different human cancer cell lines. In vitro analyses indicated that P3C strongly induced apoptosis, evidenced by accumulation of ROS, phosphatidylserine externalization, mitochondrial depolarization, caspase-3/7 activation, and cell cycle arrest in S and G2-M phases. Furthermore, P3C provoked robust microtubule disruption and a marked alteration of several kinase activities implicated in critical cell signaling pathways. Lastly, transcriptome analyses identified that P3C modulates the expression of genes involved in apoptosis, oxidative stress, kinase modulation, and microtubule stability, supporting our initial results. In the present study, we propose P3C as a powerful anticancer agent with great potential as a new anticancer therapy.

## 2. Materials and Methods

### 2.1. Cell lines and Culture Conditions

RPMI-1640 culture medium (Hyclone, Logan, UT, USA) supplemented with 100 U/mL of penicillin, and 100 µg/mL of streptomycin (Lonza, Walkersville, MD, USA) was used for the following cell lines: lymphoma-leukemia: CEM, MOLT-3, HL-60, Jurkat, NALM-6, and RAMOS; myeloma: RPMI-8226, MM1S, and MM1R; lung cancer: NCI-H358, NCI-H460 and A-549; breast carcinomas: HCC70, HCC1419, T47D; ovarian carcinomas: OVCAR-3, 5 and 8; and the prostatic carcinoma: LNCaP. The RPMI-1640 medium was supplemented with 10% heat-inactivated fetal bovine serum (FBS; Hyclone) except for HL60 and OVCAR-3, which were grown in media containing 20% FBS. Additionally, OVCAR-3 was supplemented with 0.01 mg/mL bovine insulin. Dulbecco’s Modified Eagle’s Medium (DMEM; CORNING, Corning, NY, USA), supplemented with 10% fetal bovine serum (FBS) and 100 U/mL of penicillin and 100 µg/mL of streptomycin, was used to culture the following cell lines: breast carcinomas: MDA-MB-231, MDA-MB-468, and MCF-7; pancreatic carcinoma: PANC-1; melanomas: A375 and WM-115; and the normal epithelial: Hs27 cell lines. In addition, for the MCF-7 cell line, the media were supplemented with 10 µg/mL of insulin. Furthermore, the non-cancerous breast epithelial cell line MCF10A and the prostatic adenocarcinoma PC-3 cells were cultured in DMEM F/12 media containing 10% FBS, 100 U/mL of penicillin, and 100 µg/mL of streptomycin. Additionally, the MCF10A cell line was supplemented with 10 µg/mL of insulin, 20 ng/mL of epidermal growth factor (EGF), and 0.5 µg/mL of hydrocortisone. Lastly, the ovarian adenocarcinoma cell line OV-90 was cultured in 50% of MCDB 105 medium (Sigma, St. Louis MO, USA) and 50% of Gibco medium 199 (Gibco, Waltham, MA, USA; 11150-59), complemented with 15% FBS, 100 U/mL of penicillin, and 100 µg/mL of streptomycin. All the cell lines included in this study were grown consistently at 37 °C, and in a 5% CO_2_ humidified atmosphere.

### 2.2. Differential Nuclear Staining (DNS) Assay

The DNS assay was employed to evaluate the cytotoxicity of the experimental compounds [18]. DNS assays were performed as follows: on day one 10,000 cells were seeded in 100 μL of culture media per well in a 96-well plate format (BD Falcon, Franklin Lakes NJ, USA; 353219). The next day cells were treated with increasing concentrations of the experimental compound ranging from 0.01 to 10 µM. The following control treatments were also included: vehicle (1% *v*/*v* DMSO), untreated, and positive control for cell death (1 mM H_2_O_2_). All treatments were performed in triplicate. After treatments the cells were incubated as described above for 48 or 72 h. Two hours before stopping the incubation, cells were stained with a mixture of Hoechst (Invitrogen, Waltham, MA, USA) and Propidium Iodide (PI) (MP Biomedicals, Solon, OH, USA) at a final concentration of 5 μg/mL for both dyes. In this screening strategy, the two DNA-intercalating agents, Hoechst and PI, allow the acquisition of images under two fluorescent channels, blue (Hoechst) and red (PI), to eventually determine percentages of live/dead cells. The Hoechst dye, permeable to all cells, allows quantification of whole cell populations. In contrast, PI only stains the nuclei of dead cells which have compromised cell membranes. Two hours after cells were stained, images were acquired using the IN Cell 2000 Bioimager system (GE Healthcare, Pittsburg PA, USA). A 10× objective was used and four contiguous pictures (2 × 2 image montage) were captured from each well for the Hoechst and PI fluorescent channels. Recognition of regions of interest (ROIs) on the images was performed using the IN Cell Analyzer Workstation 3.2 software (GE Healthcare) to determine the live and dead cell percentages from each well.

The cytotoxic effect of P3C was also tested after seven days of exposure to MDA-MB-231 cells. A gradient of concentrations of P3C were used (from 0.031 to 1 µM) and the following controls were included; 1% DMSO (vehicle) untreated, 100 µM H_2_O_2_, and 10 mg/mL of G418; these last two as positive controls for death. Two hours before the incubation time ended, a mixture of Hoechst and PI dyes were added to the cells and were incubated for 2 h at optimal conditions. Images from each well were acquired as described earlier in this section and percentages of live and dead cells were obtained.

### 2.3. Cell Viability Analyses

Before preparing any experimental multi-well plate, cell viability was determined using PI exclusion dye and flow cytometric assay [19]. Cell suspensions were added with 5 μg/mL PI (final concentration) and subsequently analyzed via flow cytometry (Gallios, Beckman Coulter, Miami, FL, USA). PI-positive cells were considered dead cells, and only samples with 95% or higher viability were used.

### 2.4. Cytotoxic Concentration 50% (CC_50_) and Selective Cytotoxicity Values

The CC_50_ values or cytotoxic concentration at which P3C kills 50% of the cell population were obtained by using the cytotoxicity percentage acquired from a secondary screening and linear interpolation equation (https://www.johndcook.com/interpolator.html, accessed on 10 January 2022) [20,21]. The CC_50_ and 2× CC_50_ concentrations were used in most experiments to determine the P3C dose-response effects on cells. The CC_50_ used was also determined at 24 h since several experiments were performed at that incubation period. The selective cytotoxicity index (SCI) values represent the ability of P3C to selectively kill cancer cells without causing significant damage to normal non-cancerous cells. This SCI value was calculated by dividing the CC_50_ of the non-cancerous cell line (Hs27; see Table 1) by the CC_50_ of the cancerous cell line [20,22].

### 2.5. Phosphatidylserine (PS) Distribution in Cell Membranes

The apoptosis/necrosis assay was employed to identify the mechanism of cell death induced by P3C [20]. One hundred thousand cells (MDA-MB-231) per well in 1 mL of culture media were seeded in 24-well plates. Cells were incubated overnight at 37 °C in a 5% CO_2_ atmosphere, and the following treatments were added on the next day: P3C CC_50_ and 2× CC_50_, 1% *v*/*v* DMSO (vehicle control), and 1 mM H_2_O_2_ (positive control for death). Cells were exposed to treatments for 24 h to subsequently use the Annexin V-FITC/PI kit following the manufacturer’s instructions (Beckman Coulter, Miami, FL, USA). Cells were harvested and placed in flow cytometer tubes and centrifuged for 5 min at 262× *g*. Supernatants were decanted, and 100 μL of a mixture of ice-cold 1× binding buffer containing PI and Annexin V-FITC was added to each tube by gentle resuspension. Tubes were incubated on ice in the dark for 15 min, followed by the addition of 300 μL of ice-cold 1× binding buffer. Cells were immediately analyzed by flow cytometry (Gallios, Beckman Coulter, Brea, CA, USA), collecting around 10,000 events/cells per sample. The data collection and analysis were accomplished using the Kaluza 1.3 software (Beckman Coulter).

### 2.6. Caspase-3/7 Activation Detection Assay

To monitor the activation of caspase-3/7 in live cells, the NucView 488 caspase-3/7 enzyme-substrate (Biotium, Hayward, CA, USA) and flow cytometry methods were employed. MDA-MB-231 cells were prepared and treated the same way as mentioned in the above method. Cells exposed to P3C for 8 h were subsequently collected in flow cytometry tubes. Samples were spun down (262× *g*, 5 min), and cell pellets resuspended in 500 μL of PBS containing the NucView 488 caspase-3/7 substrate at a final concentration of 5 μm. Cell samples were incubated (room temperature) in the dark for 30 min, and then 300 μL of pre-warmed PBS was added to each sample. Cells were immediately analyzed by flow cytometry. In this assay, when caspase-3/7 gets active in the cell, it will cleave the NucView 488 fluorogenic substrate, resulting in a bright green fluorescent signal. The percentages of cells exhibiting active caspase-3/7 were quantified using the FL-1 detector via flow cytometry. The Gallios flow cytometer was used for data acquisition, and 10,000 events per sample were collected. Data analysis was achieved using the Kaluza 1.3 software (Beckman Coulter).

### 2.7. Intracellular Caspase-8 Activation Assay

This assay was performed using the Caspase-8 assay kit (Abcam, Cambridge, MA, USA; ab39534) and flow cytometry analysis [23]. On day one, cells were plated (100,000 cells per well in 1 mL of growth culture media) in a 24-well plate format and incubated overnight at optimal conditions. The next day, cells were treated as mentioned previously with CC_50_ and 2× CC_50_ concentrations of P3C and the respective controls (1% DMSO, 1 mM H_2_O_2_, and untreated) for 4 h [23]. The manufacturer’s instructions were followed for staining purposes (Abcam). Cell populations emitting green fluorescence signals were identified using the FL-1 detector. Data acquisition (10,000 events) and analysis were achieved as described in Section 2.5.

### 2.8. Detection of Reactive Oxygen Species (ROS)

ROS accumulation in MDA-MB-231 cells exposed to P3C were monitored via flow cytometry using the carboxy-H_2_DCFDA (6-carboxy-2′,7′-dichlorodihydrofluorescein diacetate) fluorescein reagent [20] (Invitrogen; C400). One hundred thousand cells were seeded in 24-well plates and incubated at 37 °C in a 5% CO_2_ atmosphere for 24 h. After this step, treatments (P3C CC_50,_ P3C 2× CC_50_, 1% DMSO, or 1 mM H_2_O_2_) were added to cells and incubated for 18 h under optimal conditions. Cells were harvested and centrifuged at 262× *g* for 5 min, and cell pellets were resuspended in 1 mL of pre-warmed PBS, including the carboxy-H_2_ DCFDA reagent at a final concentration of 10 mm. Cells were incubated with the probe for 1 h at 37 °C and 5% CO_2_, then samples were spun down at 262× *g* and resuspended with 500 μL of PBS. Cells were placed at optimal conditions for 30 min to allow cell recovery and were subsequently analyzed via flow cytometry. In this experiment, when oxidation occurs, esterases modify the carboxy-H_2_ DCFDA indicator, which is initially not fluorescent, into its green fluorescent form, which allows the identification of cells undergoing oxidation and consequently accumulating ROS. For each sample, 10,000 events were obtained, and cells emitting a green fluorescent signal were gated and analyzed as described in Section 2.5.

### 2.9. Mitochondrial Depolarization Assay

To identify whether P3C induces cell death through the intrinsic apoptosis pathway, the mitochondrial membrane potential of P3C-exposed cells was evaluated by using the MitoProbe JC-1 assay kit (Molecular Probes, Eugene, OR, USA; M34152). MDA-MB-231 cells were seeded at a density of 100,000 cells in 1 mL of culture media in 24-well plates and incubated overnight at 37 °C. The following day, cells were treated with P3C CC_50_ (5.52 μm), 2× CC_50_ (11.04 μm), 1% DMSO as vehicle control, and 1 mm H_2_O_2_ as a positive control for death. Cells were incubated with P3C for 8 h at 37 °C and 5% CO_2_ atmosphere and harvested in flow cytometry tubes. Samples were centrifuged at 262× *g* for 5 min, the supernatant decanted and pellets resuspended in 500 μL of PBS containing 5 μL of the JC-1 fluorescent reagent (5′,6,6′-tetrachloro-1,1′,3,3′-tetraethylbenzimidazolylcarbocyanine iodide) and incubated 30 min at 37 °C. After incubation, 1 mL of PBS was added to each sample, and cells were spun down at 262× *g* for 5 min. Supernatants were decanted, and pellets resuspended in 500 μL of PBS, and analyzed by flow cytometry. Ten thousand events/cells per sample were acquired and analyzed as described in Section 2.5. The population of cells with a high mitochondrial membrane potential retains the JC-1 dye inside the mitochondria, forming aggregates emitting a red signal. When the mitochondria depolarize, the JC-1 aggregates become dissociated in monomers emitting a green fluorescent signal.

### 2.10. PARP-1 Cleavage Detection Assay

MDA-MB-231 cells were seeded in 96-well plates at a concentration of 10,000 cells per well in 100 µL of culture media. Cells were incubated overnight, and 5.52 µM of P3C or 1% DMSO was added to the cells. Cells were exposed to treatments for 24 h to be fixed, permeabilized, blocked, and stained. For this purpose, cells were fixed by adding formaldehyde at a final concentration of 4%. Samples were incubated for 20 min at room temperature, and then formaldehyde was removed from each well, and cells were washed and permeabilized twice with 200 µL of 0.1% Tween 20 in PBS for 10 min each at room temperature. Next, 200 µL of a blocking solution (5% bovine serum albumin in TBS-T; Tris-buffered saline with 0.1% Tween 20) were added to each well, and plates were incubated for 1 h on a rocking platform at room temperature. Afterward, the blocking solution was removed from each well. Cells were stained with the following mixture for 1 h at room temperature: 50 µL of 0.1% Tween 20 in PBS, 5 µg/mL of DAPI, 0.165 µM of phalloidin conjugated to Alexa Fluor 568, and 0.5 µg/mL of an anti-tubulin monoclonal antibody conjugated to Alexa Fluor-488 (Thermo Fisher Scientific, Rochester, NY, USA). Cells were washed three times with 200 µL of 0.1% Tween 20-PBS solution, and 200 µL of this solution was left in each well for imaging purposes [24]. Images were acquired and analyzed by using the LSM-700 confocal microscope and Zen 2009 6.0 software (Zeiss, White Plains, NY, USA).

### 2.11. Analysis of the Cell Cycle Progression

The potential effect of P3C on the cell cycle profile was investigated by measuring the total cellular DNA content via flow cytometry as previously described [22]. Asynchronous MDA-MB-231 cells were seeded in a 24-well plate format as described above and treated for 72 h with low concentrations of P3C (CC_10_ and CC_20_). The strategy of using low concentrations of the experimental compound is to prevent excessive amounts of DNA fragmentation or cell death, which will allow the detection of arrest at any of the cell cycle phases. Control treatments were also included, as mentioned above. Cells were harvested, centrifuged, and cell pellets were resuspended with 100 μL of culture media. Afterward, 200 μL of nuclear isolation media (NIM)-DAPI solution (Beckman Coulter) was added to each sample, and readings were acquired via Gallios flow cytometer (Beckman Coulter). The NIM-DAPI solution fixates, permeabilizes, and stains the cell nuclei, allowing quantification of the total DNA content of each cell. For these experiments, 100,000 events were acquired to obtain a well-defined cell cycle distribution profile. The FL9 detector was used to capture the DAPI-DNA emitting fluorescent signal (~461 nm). This approach does not require cellular RNA removal due to the longer-wavelength fluorescence signal emitted by the DAPI-RNA complexes (~500 nm).

### 2.12. Scratch-Wound Assay to Determine Cell Metastatic/Invasion Activity Using Live-Cell Microscopy

The scratch wound-healing strategy was used to measure cell migration/invasion inhibition, interpreted as an antimetastatic activity. Here, we introduce a novel approach combining nuclear fluorescence staining (blue; Hoechst) with brightfield illumination channels to increase accuracy in counting the wounded zone cells. Migration of MDA-MB-231 breast cancer cells was performed as previously described with additional modifications [25]. Briefly, 50,000 cells were seeded per well into a 24-well plate and cultured until forming a confluent monolayer, around three to four days. Afterward, a scratch was created with a sterile pipette tip (1 to 200 µL volume; VWR, Cat. No. 53508-810) on the confluent cell monolayer, thus, making a cell-free gap (wound). Next, wells were washed with fresh media to remove detached cells and then fresh culture media containing 5% FBS. Subsequently, cells were exposed to 5.52 µM of P3C (CC_50_) and incubated for 24 h. One hour before the 24 h incubation period, cells were exposed to 5 µg/mL of Hoechst 33,342 to stain the nuclei. Cell migration to the cell-free wounded area was examined at 0 and 24 h by capturing live-cell confocal microscopic digital images (LSM 700, Zeiss), brightfield illumination (phase contrast), and blue channels (Hoechst), by using a 10× objective, 405 nm laser and 1 Airy-Unit (AU) pinhole settings. For the 0 h image preparation time, cells were incubated with 5 µg/mL of Hoechst for 1 h before capturing pictures. Moreover, the time involved in capturing images was short since both fluorescence and brightfield images were acquired simultaneously; and neither fixation nor additional washes were necessary. To define the zone where the migration of the cells was counted, a demarcated rectangular surface area was created inside the cell-free corridor at time 0 h, equidistantly to the borders of the wounded area; the size of this rectangular area was maintained consistently throughout all the analyzed samples (Appendix A). After 24 h of incubation, the cell numbers within the wounded-scratched area and located inside the rectangular demarcated regions were manually counted. For each experimental point, 10 independent counts were performed. Two-tailed paired Student’s *t*-tests were used to determine the statistical significance among the two sample groups.

### 2.13. Cytoskeleton Analysis via Confocal Immunofluorescent Microscopy

For this series of experiments, MDA-MB-231 and HeLa cells were utilized. The 2500 cells in 100 µL of complete media were seeded in 96-well tissue culture-treated imaging microplates (BD Falcon, 353216). Cells were incubated overnight, and the following day treatments were added as follows: P3C CC_50_ (5.52 µM), Paclitaxel (1 µM), and Cytochalasin D (5 µg/mL), the last two being controls of microtubule disruption and actin polymerization inhibition, respectively. As controls, 1% DMSO and H_2_O_2_ (1 mM) were also included. Cells were exposed to treatments for 2 h under optimal conditions. Then, cells were fixed by directly adding 100 µL of fresh 8% formaldehyde without removing any media, reaching a 4% formaldehyde concentration. Plates were incubated for 20 min at room temperature, and formaldehyde was removed from each well to subsequently wash and permeabilized with 200 µL of 0.1% Tween 20 detergent in PBS for 10 min at room temperature. Washes were repeated two more times for each well without an incubation period. After the washes, the Tween 20-PBS solution was removed, and 200 µL of a blocking solution (5% bovine serum albumin in TBS-T; Tris-buffered saline with 0.1% Tween 20) was added to each well. Plates were incubated for 1 h on a rocking platform at room temperature. After removing the blocking solution from each well, cells were stained for 1 h on a rocking platform at room temperature using the following: 50 µL of PBS solution containing 0.1% Tween 20, DAPI (5 µg/mL), phalloidin conjugated to Alexa Fluor 568 (0.165 µM), and an anti-tubulin monoclonal antibody conjugated to Alexa Fluor-488 (Thermo Fisher Scientific). Three consecutive washes were completed for each well with 200 µL of 0.1% Tween 20-PBS solution and 200 µL of this solution was left in each well for imaging. Immunofluorescent images were captured with a laser scanning confocal microscope (LSM-700, Zeiss) using an EC Plan-Neofluar 40×/1.30 oil DIC objective and assisted with Zen 2009 6.0 software (Zeiss).

### 2.14. Kinase Multiplex Luminex Assays

MDA-MB-231 cells (5 × 10^6^ per sample) were collected in microcentrifuge tubes (1.5 mL) and treated for 3 h at 37 °C with 0.1 and 10 µM of P3C. Tubes were gently flicked every 10 min until incubation time ended. Afterward, cells were centrifuged at 262× *g* for 5 min, and cell pellets were lysed with 50 µL of Milliplex MAP cell-signaling lysis buffer containing a protease inhibitor cocktail (Millipore, Billerica, MA, USA). Cell lysates were incubated under constant rotation for 1 h at 4 °C and then centrifuged for 14,000 rpm for 15 min at 4 °C. The cleared supernatants were collected and retrieved in fresh tubes. Protein concentrations were determined by using the NanoDrop UV-Vis spectrophotometer (Thermo Fisher Scientific). For each sample, 25 µg of total protein from the cell lysate was used following the vendor’s recommendations (Millipore). The Milliplex MAP human multi-pathway signaling network kit (Millipore) was used to detect the levels of phosphorylation/activation of the following proteins: CREB (pS133), ERK (pT185/pY187), NF-kB (pS536), JNK (pT183/pY185), p38 (pT180/pY182) and STAT3 (pS727). In addition, the Milliplex MAP 8-Plex Human Src Family Kinase Kit (Millipore, Billerica, MA) was also implemented for the analysis of activation of the following kinases: Blk (Tyr389), Fgr (Tyr412), Fyn (Tyr420), Hck (Tyr411), Lck (Tyr394), Lyn (Tyr397), Src (Tyr419), Yes (Tyr421). Data acquirement and analysis were accomplished via xPONENT 3.1 software (Luminex, Austin, TX, USA).

### 2.15. Whole Transcriptome Analysis via Next-Generation Sequencing

MDA-MB-231 and MDA-MB-468 breast cancer cells (1,000,000 cells in 5 mL of culture media per treatment in 24-well plates) were treated with P3C 2× CC_50_ (calculated at 24 h of exposure) or vehicle control for 6 h and collected for RNA extraction. The RNeasy Mini Kit (Qiagen; Germantown, MD, USA; 74104) was used following the manufacturer’s RNA isolation instructions. A NanoDrop spectrophotometer (Thermo Fisher; ND-ONE-W) was utilized to verify the RNA quantity and purity. Only samples that fell into the RNA purity A260/A280 ratio between 1.8 and 2.1 were considered for further analysis. Before library preparation, the concentrations of the RNA samples were quantified via RNA BR Assay Kit for Qubit 3.0 (Invitrogen; Q10210) and integrity (RINe > 7.0) with RNA Screen Tape (Agilent; Santa Clara, CA, USA; 5067-5576) in a 2200 TapeStation (Agilent; G2964AA). The TruSeq Stranded mRNA library prep kit (Illumina, San Diego, CA, USA; 20020594) was used to convert the mRNA into a final cDNA library. Next, the prepared cDNA was sequenced using the NextSeq 500 High Output Kit v2.5 (Illumina; I20024907) following the manufacturer’s instructions, and the NextSeq 500 system (Illumina). A total of three biological replicates were included for each cell line and each cell treatment.

### 2.16. RNA-seq Data Analysis

For mRNA-seq data visualization and analysis, we utilized pipelines that integrated the QC (FastQC, ShortRead), trimming process (Trimmomatic-v0.36), alignment (Tophat2), reads quantification (Cufflinks), and differentially expressed gene (DEG) analysis (Cuffdiff). Briefly, the RNA-seq raw FASTQ data were first trimmed using Trimmomatic (V0.36). The trimmed reads were aligned to the human reference genome (NCBI GRCh38) with TopHat V2.1.1 using default parameter settings. The aligned bam files were then processed using Cufflinks V2.2.1 for gene quantification. Genes with FPKM ≥ 1 in all samples were used to identify Differentially Expressed Genes (DEGs). DEGs were identified by Cuffdiff using a cutoff of > 0.05 for FDR and > 2 for fold change (FC), respectively. Hierarchical clustering was used to generate a heat map of all DEGs from all cell lines and drug treatments. Clustering was done with “R” (http://cran.r-project.org, accessed on 10 January 2022), using average linkage and Spearman Rank Correlation as clustering and distance measurement methods, respectively.

### 2.17. Gene Ontology (GO), and Ingenuity Pathway Analyses (IPA)

Gene ontology (GO) enrichment analysis of the 28 DEGs found in common between the two TNBC cell lines tested was implemented using the STRING database (https://string-db.org, accessed on 10 January 2022). The GO terms for biological processes with corrected *p*-values ≤ 0.05 (shown as false discovery rates) were considered significant. The top 30 GO terms for biological processes were selected for further analysis. Moreover, the Ingenuity pathway analysis was used to analyze the DEGs. IPA is built on a comprehensive, manually curated content of the QIAGEN Knowledge Base, which, along with powerful algorithms, helps in the identification of the most significant pathways and causal relationships associated with experimental data. Tools for causal analytics that are available within the IPA software, (QIAGEN’s Ingenuity^®^ Pathway Analysis software, build version 389077M, QIAGEN, Redwood City, CA, USA) like ‘Upstream Regulator Analysis’, ‘Causal Network Analysis’, and ‘Downstream Effects Analysis’ were used to analyze the said DEGs.

### 2.18. Statistical Analyses

Each experimental point denotes at least three independent measurements, except where otherwise noted. The results are shown as the average of the several measurements with their respective standard deviations to indicate the experimental variability. The *p*-values were calculated using a two-tailed paired Student’s *t*-test to establish statistical significance between two samples. In some occasions, the significant *p*-values (<0.05) were designated with asterisks; * *p* < 0.05, ** *p* < 0.01, and *** *p* < 0.001.

## 3. Results

### 3.1. Drug Screening Identified a Compound with Potent Cytotoxic Activity on a Panel of 27 Human Cancer Cell Lines Derived from Different Tissues

A chemical library of 10,000 drug-like small molecules (ChemBridge DIVERSet, San Diego, CA, USA) was used to identify compounds with cytotoxic potential against human cancer cells via a high-throughput screening initiative. P3C was initially identified as cytotoxic to MDA-MB-231 cells (Figure 1) in a primary screening of 4640 distinct compounds using the Differential Nuclear Staining (DNS) assay^17^. P3C was the most cytotoxic of the screened compounds with a CC_50_ in the nanomolar range (0.49 µM; Figure 2). As shown in Table 1, 28 different human cell lines were subsequently analyzed via the DNS assay (27 of cancer origin and one non-cancerous cell line), and the CC_50_s were obtained using both experimental values and linear interpolation. Due to a slower replication rate, all adherent cell lines were incubated for 72 h while the non-adherent cell lines were examined after 48 h of exposure to P3C^19^. In general, most of the cell lines tested demonstrated a cytotoxic concentration (CC_50_) below 1 µM. The most sensitive adherent cell lines to P3C were the NCI-H358 non-small cell lung cancer and the MDA-MB-468 breast cancer cells with CC_50_ values of 0.19 and 0.25 µM, respectively. Of the rapidly dividing non-adherent cell lines, the most sensitive to P3C was the RAMOS Burkitt’s B lymphoma and the Jurkat Acute T-cell Leukemia cell lines with CC_50_ values of 0.31 and 0.37 μm, respectively. Interestingly, the MOLT-3 acute lymphoblastic leukemia and the HCC1419 breast carcinoma cell lines were the most resistant to P3C, with CC_50_s of 6.54 and 6.27 µM, respectively (Table 1). To determine cancer-type selectivity, we selected the CC_50_ from the P3C-treated non-cancerous Hs27 human fibroblast cell line to calculate the selective cytotoxicity index (SCI). Table 1 depicts the SCI of the P3C for a panel of cancer cells compared with non-cancerous cells^21^. As shown in Table 1, P3C displayed significant selectivity against various tumor types, including the MDA-MB-231 (SCI = 7.1) and another TNBC cell line MDA-MB-468 (SCI = 13.9). However, this selectivity varied significantly with other breast cancer lines, ranging from 0.6 for HCC70 and HCC1419 to 4.5 and 7.7 for MCF-7 and T47D, respectively. Interestingly the highest SCI values were obtained with the Burkitt’s lymphoma cell line RAMOS with an SCI of 18.6 (Table 1). Given the range of CC_50_ and SCI values from 0.19 µM (SCI = 18.3) to 6.54 µM (SCI = 0.9) for MOLT-3, there was no clear pattern of cancer type selectivity, perhaps indicating that P3C acts on cells expressing or lacking the expression of specific genes or pathways (see Discussion). P3C had a significantly low CC_50_ of 0.49 µM on triple-negative MDA-MB-231 cells, and therefore this cell line was selected to characterize the mode of action of P3C. In addition, the MDA-MB-231 cell line is considered a prototype of triple-negative breast cancer cells and is frequently used in breast cancer research.

Additionally, a long exposure analysis of P3C cytotoxicity was also performed in MDA-MB-231 cells. A gradient of concentrations of P3C (from 0.031 to 1 µM) was added to cells, and cytotoxicity was assessed after 7 days of treatment. Results demonstrated a consistent P3C cytotoxicity in which the potency of the compound is maintained from 72 h (CC_50_ 0.49 µM) to 7 days (CC_50_ 0.41 µM; Appendix A).

### 3.2. P3C Induces Cell Death via Apoptosis as Determined by PS Externalization and Caspase Activation

The mechanism of cell death induction by P3C was investigated by measuring phosphatidylserine (PS) externalization through flow cytometry after 24 h of exposure to MDA-MB-231 cells. P3C’s CC_50_ (5.52 µM) and twice the CC_50_ (11.04 µM) were used for these analyses. Our data show a significant increase in apoptosis on cells treated with P3C when compared to vehicle-control-treated cells (1% DMSO, *p* < 0.01, Figure 3A and Appendix A). Untreated and H_2_O_2_-treated samples were also included as negative and positive controls, respectively. Cells that displayed PS externalization were identified as early apoptotic (Annexin V-FITC positive), and cells with compromised plasmatic membranes (Propidium iodide permeable) and/or with PS externalization, were cataloged as late apoptotic cells. Percentages shown in Figure 3A represent total apoptosis values resulting from the sum of both early and late apoptosis (Appendix A).

Caspase-3 is a central mediator of apoptosis, activated in response to apoptotic stimuli, crucial for some characteristic nuclear and morphological changes in the cell, such as apoptotic DNA condensation and DNA fragmentation [26]. To further investigate the cell death mechanism induced by P3C, caspase-3/7 activation was evaluated on MDA-MB-231 cells upon exposure to P3C. Cells were treated with 5.52 μm and 11.04 μm of the compound and incubated for 8 h. Cells emitting a green fluorescence signal due to caspase-3/7 activation were monitored via flow cytometry (FL1 detector; see Materials and Methods for details). Our results confirmed significant caspase-3/7 activation induced by P3C, compared to untreated and solvent control (*p* < 0.001, Figure 3B and Appendix A), indicating that P3C induces apoptosis via activation of caspase-3/7 in MDA-MB-231 cells.

Caspase-8 is a key activator of the extrinsic apoptotic pathway, and its activation is initiated as a response to the ligation of death receptors [27]. To investigate whether caspase-8 is involved in P3C-mediated cell death induction, we examined caspase-8 activation in MDA-MB-231 cells upon exposure to P3C for 4 h. Cells with activated caspase-8 were analyzed by flow cytometry. P3C at twice the CC_50_ concentration, 11.04 μM, triggered a slight but significant caspase-8 activation after 4 h of exposure (*p* < 0.01, Figure 3C and Appendix A) in MDA-MB-231 cells. Unlike what was observed with activation of caspase-3/7, P3C at a lower concentration of 5.52 µM (CC_50_) did not induce cleavage, suggesting that P3C can activate the extrinsic apoptosis pathway at high concentrations. Furthermore, as expected, toxicity caused by H_2_O_2_ (1 mM) provoked high levels of caspase-8 activation, whereas solvent (DMSO) and untreated controls did not show activation (Figure 3C and Appendix A). Our results demonstrate that P3C activated the extrinsic pathway of apoptosis when a high concentration of P3C was used on MDA-MD231 cells.

### 3.3. Induction of Reactive Oxygen Species (ROS) and Mitochondrial Depolarization by P3C

An excess of reactive oxygen species in the cell has detrimental effects on nucleic acids, proteins, lipids, membranes, and organelles such as mitochondria, which leads to apoptosis [28]. To further characterize the harmful impacts of P3C that result in programmed-cell death, MDA-MB-231 cells were assessed for ROS production after 18 h of treatment with P3C. P3C-treated cells were stained with the carboxy-H2DCFDA fluorescein reagent for 1 h. Only cells undergoing oxidative stress converted the carboxy-H2DCFDA reagent into its green fluorescent form, which was subsequently detected by flow cytometry. After treatment with the CC_50_ and twice the CC_50_ concentrations of P3C, 56.6% and 54.14% of the cells demonstrated ROS accumulation, respectively (Figure 4A and Appendix A).

Healthy mitochondria require an electrical potential across the inner membrane termed mitochondrial transmembrane potential (ΔΨm). A loss in ΔΨm is a critical occurrence in the intrinsic apoptosis pathway, and ROS overproduction is associated with loss of mitochondrial membrane potential [29,30]. To investigate if P3C alters mitochondrial health and contributes to apoptosis, we examined the ΔΨm of P3C-exposed MDA-MB-231 cells for 8 h. P3C-treated cells were stained with the polychromatic fluorescent reagent JC-1 and analyzed via flow cytometry. Upon a loss of ΔΨm, intracellularly, the JC-1 fluorescence emission signal shifts from red to green, and both are detected using FL2 and FL1 detectors, respectively. As shown in Figure 4B and Appendix A, MDA-MB-231 cells exposed to either the P3C at the CC_50_ or twice the CC_50_ exhibited a significant amount of mitochondrial depolarization when compared to the vehicle (DMSO) or untreated controls (*p* < 0.01). Additionally, as expected, H_2_O_2_ treatment resulted in a significant percentage of cells with depolarized mitochondria. Our results indicate that P3C induces both ROS overproduction leading to mitochondrial depolarization and activation of the intrinsic apoptotic pathway. 

### 3.4. P3C Induces PARP Cleavage in MDA-MB-231 Cells

PARP, a 116 kDa nuclear poly (ADP-ribose) polymerase, is mainly cleaved by caspase-3/7 in vivo, separating the amino-terminal DNA binding domain (24 kDa) from the carboxy-terminal catalytic domain (89 kDa), which is a biochemical hallmark of cells undergoing apoptosis [31,32]. To further support and extend our findings on the mechanism of cell death induced by P3C, PARP cleavage was evaluated in MDA-MB-231 cells after being exposed for 24 h to P3C. Cells that were stained with DAPI, phalloidin-Alexa 568, and α-cleaved PARP-Alexa 488 antibodies (recognizes the 89 kDa PARP fragment) were detected by confocal microscopy. Results demonstrate that P3C triggers PARP cleavage, as evidenced by cells emitting a green signal (Appendix A). In contrast, the vehicle control (DMSO)-treated cells showed no PARP cleavage (Appendix A), with no detected green signal, since the antibody used only recognizes cleaved PARP. Our results further support the prior results that once P3C activates caspase-3/7 it leads to PARP cleavage, which is one of its downstream substrates [33].

### 3.5. P3C Disrupts Cell Cycle Progression by Arresting Cells in S and G2-M Phases

To further characterize the P3C effect on MDA-MB-231 cells, an analysis of the cell cycle progression was performed after exposing the cells for 72 h with low concentrations of P3C (CC_10_ = 1.1 µM and CC_20_ = 2.2 µM). It is important to note that by using the CC_50_ or higher concentrations in this assay, excessive DNA fragmentation of cell subpopulations (sub-G0-G1; hypodiploid) is seen, which negatively interferes with the definition of the cell cycle phases (gates). After staining cells with the nuclear isolation media containing DAPI (NIM-DAPI), the DAPI-DNA signal intensity was used to quantify total cellular DNA content through flow cytometry. A significant cell cycle arrest was observed in the S (hyperdiploid) and G2-M (tetraploid) phases on cells exposed to low concentrations of P3C (CC_10_ or CC_20_) when compared to the solvent control (DMSO; *p* < 0.001, Figure 5C,D). As might be expected from cells undergoing nuclear DNA fragmentation, an increase in the sub-G0-G1 hypodiploid phase was detected (Figure 5A; *** *p* < 0.001, and ** *p* < 0.01). In addition, a robust reduction of the diploid G0-G1 phase was also found, most likely as a consequence of the increased percentages in the S and G2-M phases (Figure 5B, *p* < 0.001). As in prior assays, solvent (1% DMSO), untreated (Unt), and H_2_O_2_-treated cells were included as controls. Statistical analyses were made comparing experimental treatments against solvent control treatment. Hence, our data suggest that P3C alters the normal cell cycle progression by arresting MDA-MB-231 cells in S and G2-M phases and increasing the sub-G0-G1 subpopulation, identified as the apoptotic subpopulation undergoing DNA fragmentation.

### 3.6. P3C Reduced Cell Proliferation by Inhibiting Cell Migration/Invasion Activity

The potential effect of P3C on the migration of MDA-MB-231 cells in vitro was also explored. Cell-free gaps were created by making a scratch using a sterile pipette tip on a confluent monolayer of cells. After control and compound treatments, the total number of cells migrating into the cell-free gaps was manually quantified. As expected, solvent control (DMSO-treated) cells exhibited no inhibition of migration into the cell-free wounded zone, and these were used as a reference for the highest migration rate of 100%. In contrast, P3C exerted a potent and significant (*p* < 0.00001) inhibitory effect on MDA-MB-231 cell migration, as compared to the solvent control, which resulted in 6.6 ± 3.15 and 125 ± 24.45 cells in the demarcated region in the gap, respectively; resulting in a decrease of 94.72% of cell migration (Appendix A). In addition, P3C was more efficient than doxorubicin in reducing the number of cells by 85.12% migrating to the demarcated region (or 18.6 ± 11.2 cells counted; Appendix A). So far, these findings suggest that P3C has potential as an anti-metastatic drug by inhibiting cellular migration.

### 3.7. P3C Treatment Leads to Microtubule Structure Disruption

Analysis of the potential effects of P3C on the cytoskeleton was explored by assessing both microfilament (F-actin) and microtubule (tubulin) integrity by confocal microscopy after treating MDA-MB-231 and HeLa cells for 2 h with P3C. Cells treated with P3C, Paclitaxel, cytochalasin D, DMSO, and untreated samples were included as controls. Cells were fixed, permeabilized, and immunostained with α-tubulin-Alexa-488 antibodies (microtubules), phalloidin-Alexa-568 (F-actin), and DAPI (nucleus) to subsequently capture digital confocal microscopy images [24,34]. Several images were acquired from each treatment and the morphological changes of both microfilaments and microtubules were meticulously inspected. As shown in Figure 6E and Appendix A, the tubular arrangement of microtubules disappeared in both cell lines after 2 h of P3C treatment, compared to DMSO and untreated controls. Nonetheless, the Alexa-488 green fluorescent signal (microtubules) was detected as a punctuated pattern throughout the cytoplasm of HeLa cells and on the periphery of MDA-MB-231 cells. On the other hand, the actin fibers or microfilaments appear to be unchanged after P3C exposure in both cell lines (on the Alexa-568 channel; Figure 6E and Appendix A) when compared to the vehicle and untreated controls. This phenotype is different from the pattern shown with cytochalasin D, a potent inhibitor of actin polymerization (Figure 6C and Appendix A). Microtubule disruption mediated by P3C seems to be more pronounced than in the Paclitaxel control (Figure 6D,E and Appendix A), an inhibitor of the mitotic spindle and suppressor of microtubule dynamics [35]. Microtubules are barely observed in the two P3C-treated cell lines tested (Figure 6E and Appendix A). The DMSO and untreated controls consistently unveiled an unaffected cytoskeleton organization in the two cell lines tested (Figure 6A,B and Appendix A). These results indicate that P3C acts as a microtubule-disrupting drug and could explain the loss of migration of cells after treatment.

### 3.8. P3C Perturbs and Inhibits Microtubule Mitotic Spindle Formation in HeLa Cells

To confirm and extend the disruptive effects of P3C on microtubules, HeLa cells were also exposed for 2 h to P3C, cytochalasin D, Paclitaxel, and DMSO. After treated cells were fixed, permeabilized, and stained as described previously, images of the selected cells were acquired of cells in metaphase/anaphase stages of mitosis. In these assays, HeLa cells were used to facilitate the study of the structural organization of the microtubules during mitosis since HeLa cells present a higher number of cells undergoing cell division on a microscopic field of view compared to other cell lines. Due to the flattened shape of these cells, both the different mitosis phases and the mitotic spindle can be easily recognized. Representative images obtained from each treatment are depicted in Appendix A. Untreated and solvent-treated controls show healthy cells undergoing mitosis in the anaphase and metaphase stages, respectively (Appendix A), as evidenced by the microtubule filamentous shape forming the mitotic spindle (in the Alexa-488 (green) channel) and the condensed DNA comprising the chromosomes (seen in the DAPI (blue) channel). Cytochalasin D treatment also displayed an unaltered microtubule spindle formation and intact chromosome alignment of a cell experiencing mitosis in the metaphase stage (Appendix A). Furthermore, as expected, the actin filaments were completely abrogated and displaced, forming actin aggresomes in a punctuated form (Appendix A). On the contrary, Paclitaxel treatment showed an unchanged microfilament actin structure and disorganized microtubule spindles that cannot sustain chromosome aggregation, resulting in a disordered arrangement (Appendix A). Lastly, and most importantly, cells treated with P3C had a completely distorted microtubule organization, with no noticeable tubulin fibers or spindle (see the green dot pattern in Appendix A). Although the cell has already started cell division, the lack of a functional mitotic spindle cannot maintain chromosome organization. Thus, the cell cycle gets interrupted in the early stages of mitosis. Consistent with these findings, from several images and cells analyzed for this treatment, no cells were detected in later mitosis stages. Conversely, P3C treatment did not affect the typical actin organization in mitosis, in which the microfilaments aggregate at the cell periphery, forming what is called the actomyosin cortex (Appendix A). Taken together, the data demonstrate that P3C adversely affects microtubule organization, leading to defective mitotic spindle formation unable to accomplish cell division; thus, P3C causes a mitotic catastrophe, in which a cell cannot complete mitosis successfully due to insufficient chromosome condensation and segregation, leading to cell death [36]. This microtubule disruption and mitotic anomaly caused by P3C is congruent with the G2/M arrest and DNA fragmentation detected during the cell cycle analysis.

### 3.9. P3C Decreased CREB, p38, ERK, Akt, P70S6K, and STAT3 and Increased JNK and NF-kB Phosphorylation in MDA-MB-231 Cells

Cell signaling analyses were performed in P3C-treated MDA-MB-231 cells to determine whether specific signaling pathways are induced or inhibited during the P3C cell death. A panel of kinases, transcription factors, and adaptor proteins implicated in various signaling pathways were examined using the bead-based Multiplex immunoassays and Luminex technology. Cells were exposed for 3 h to P3C, and total protein extracts were used to measure the phosphorylation/activation levels of p38, ERK1/2, CREB, STAT3, STAT5A/B, JNK, NF-kB, Akt, and p70 S6K. For this analysis, an antibody/bead-based multiplex assay that distinguishes specific phosphorylation sites within p38 (pT180/pY182), ERK1/2 (pT185/pY187), JNK (pT183/pY185), CREB (pS133), STAT3 (pS727), STAT5A/B (pY694/699), NF-kB (pS536), Akt (pS473), and p70 S6K (pT412) was utilized (Figure 7 and Appendix A). Out of these nine effector proteins, P3C-treatment resulted in a significant decrease in the phosphorylation of p38, ERK1/2, CREB, and STAT3 (*p* < 0.03; 50% or less; Figure 7A,B,D,E). In contrast, under the same conditions, P3C increased the JNK and NF-kB phosphorylation levels by more than two-fold (*p* < 0.03; Figure 7C,F). STAT5 phosphorylation values were below the detection level in MDA-MB-231 cells under our experimental conditions (data not shown). In addition, both Akt and P70S6K kinases were partially inactivated; however, the *p*-values were not significant (Appendix A). Moreover, all of the P3C down- and up-regulated phosphorylation effects were in a dose-dependent manner. As expected, the most significant changes detected were seen when cells were exposed to the highest concentration of P3C tested (10 µM; Figure 7). These assays revealed that P3C elicited down-regulation of the p38MAPK/STAT3, ERK1/2/CREB, and PI3K/Akt pathways and suggested a complex interaction of proteins after P3C-induced programmed cell death.

### 3.10. P3C Treatment Resulted in Fyn Tyrosine Kinase Dephosphorylation

Members of the Src tyrosine kinase family involved in cell signaling were assessed to elucidate whether those enzymes were implicated in the P3C-mediated cytotoxicity. MDA-MB-231 cells were treated for 3 h with a concentration gradient of P3C, and protein extracts were used to examine Fyn and Src’s activation. A bead-based multiplex technology and antibodies reacting with specific tyrosine phosphorylation sites, Fyn (Tyr420) and Src (Tyr419), were used. Fyn was found to be significantly dephosphorylated (~50% lower; *p* = 0.0058) after 0.1 µM of P3C exposure (Figure 8A). However, when cells were treated with 1 µM of P3C, the Fyn dephosphorylation was more accentuated (~62% reduced; *p* = 0.0034), as compared with solvent control cells (Figure 8A). Moreover, the Src kinase exhibited a slight non-significant decrease in phosphorylation after P3C treatment (Figure 8B). Thus, our data demonstrate that P3C reduced Fyn activity in TNBC MDA-MB-231 cells in a concentration-dependent manner.

### 3.11. Transcriptome Analysis of P3C-Treated MDA-MB-231 and MDA-MB-468 Cells Revealed 28 Differentially Expressed Genes in Common

To determine the expression of genes affected by P3C treatment, we analyzed the gene expression profiles from P3C-treated triple-negative breast cancer cells (TNBC) MDA-MB-231 and MDA-MB-468 via transcriptome sequencing analyses (RNA-seq; Figure 9). To select the genes that were differentially expressed (DEGs) in both P3C-treated MDA-MB-231 and MDA-MB-468 cells, we compared the gene expression profiles to that of solvent/vehicle-treated control cells with a cut-off of ≥2.0 or ≤−2.0-fold change for up-and down-regulated genes, respectively. P3C-treated MDA-MB-231, and MDA-MB-468 cells revealed 177 and 472 genes that were significantly up-regulated (Figure 9 and Figure 10A), whereas 249 and 89 genes that were significantly down-regulated, respectively (Figure 9 and Figure 10B). Further analyses of the DEGs from MDA-MB-231 and MDA-MB-468 cells led to 28 genes that were differentially expressed in both cell lines with 23 up-regulated and 5 down-regulated genes (Figure 10A–C). Interestingly, although two TNBC cancer cell lines were analyzed, the basal levels of gene expressions without treatment (PEG/PBS solvent controls) were quite diverse between each other (Figure 9A,C). In summary, our data revealed 23 up-, and 5 down-regulated genes induced by P3C in the two TNBC cell lines tested, suggesting an interesting common mode of action that ultimately leads to antiproliferative and pro-apoptotic effects.

### 3.12. In P3C-Treated MDA-MB-231 and MDA-MB-468 Cells, the Common DEGs Are Implicated in Apoptosis, Stress Response, MAPK Kinases Inactivation, and Microtubule Structure and Stability

Transcriptome (RNA-seq) analysis, integrated with bioinformatics and in vitro assays, revealed potential genes and pathways associated with P3C-induced antiproliferative and apoptotic activities on two TNBC cell lines. The 28 DEGs found in common in P3C-treated MDA-MB-231 and MDA-MB-468 cells were subjected to Gene Ontology (GO) and Ingenuity pathway analyses to investigate the biological processes and pathways that these genes were associated with. The top 30 GO terms found for the 23 up-regulated genes were sorted by significance (corrected *p*-values ≤ 0.05) and are presented in Appendix A. In addition, the biological processes that were considered relevant for this study are shown highlighted in Appendix A. In general, the relevant up-regulated genes are involved in the regulation of apoptosis, control of cell proliferation, response to stress, and regulation of protein phosphorylation, more specifically, inactivation of MAPK function. For instance, Dual Specificity Phosphatase 4 and 10 (*DUSP4* and *DUSP10*) genes that were up-regulated in P3C-treated cells encode protein phosphatases that are essential regulators of the oxidative stress response that inactivate MAP kinases (ERK1/2, p38, JNK) by dephosphorylation. Consequently, the overexpression seen on both genes can be directly associated with our prior results indicating that P3C down-regulates p38 and ERK1/2 MAP kinases via their dephosphorylation. Moreover, the *BTG2* gene encodes a protein involved in antiproliferative, antimetastatic, anti-invasive responses as a cell cycle regulator, and possesses mitochondrial depolarization activity. This *BTG2* gene was also found to be up-regulated by P3C in this study, and its overexpression can contribute towards the observed abrogated cell migration and could also have promoted mitochondrial depolarization. On the other hand, no significant GO terms were found to be associated with the five down-regulated genes, which may be due to the small number of DEGs. Nonetheless, it is relevant to mention that the Proline and Serine Rich Coiled-Coil 1 gene (*PSRC1*) was down-regulated, and it has a critical role in microtubule maintenance and stability and is required for successful mitosis. Additionally, *PSRC1* regulates the mitotic spindle dynamics and controls the turnover rate of microtubules on metaphase spindles [37,38]. *PSRC1* downregulation supports observation by immunocytochemistry and microscopy that microtubules were found disorganized, with a deformed spindle that was unable to complete mitosis on dividing cells upon P3C treatment (Figure 6E, Appendix A). Another interesting finding was the up-regulation of the N-Myc Downstream Regulated 1 (*NDRG1*) gene that protects cells from spindle disruption damage [39]. The overexpression of this gene could have resulted as a cellular response to microtubule impairment caused after exposure to P3C.

Ingenuity pathway analyses were also used to investigate the pathways associated with the 28 DEGs found in common in the two TNBC cell lines tested. These analyses revealed several canonical pathways (Appendix A); the most relevant for this study are listed in Table 2. For instance, the SAPK/JNK signaling pathway is regulated by *DUSP4* and *DUSP10* and regulation of cell cycle progression by the *BTG2* gene. Additionally, *DUSP10* was also found to be involved in the p38 MAPK signaling pathway, *PIM1* on the STAT3 pathway, and *DUSP4* on ERK/MAPK signaling pathways (Table 2).

## 4. Discussion

Cancer is a significant public health and economic burden worldwide and the second leading cause of death globally, only secondary to cardiovascular disease [1]. One in five men and one in six women worldwide develop cancer during their lifetime [40]. In addition, one in eight men and one in eleven women die from this disease [40].

After an extensive high-throughput screening for anticancer drugs, a pyrazole 3-carbohydrazide (P3C) compound was identified as a potential antineoplastic agent, exerting potent cytotoxicity to 27 human cancer cell lines from diverse tissue origins. This compound was also found to have weak activity against a non-cancerous cell line that was included for comparison purposes. The majority of secondary assays were performed on the TNBC MDA-MB-231 cell line derived from a highly aggressive, invasive, and poorly differentiated cell type, one of the most frequently diagnosed malignancies, with poor prognosis and the leading cause of cancer-related death among women worldwide [41]. 

Existing cancer therapies often demonstrate their antitumor activity by promoting apoptosis in cancer cells, generally characterized by distinctive morphological and biochemical hallmarks such as chromatin condensation, DNA fragmentation, and phosphatidylserine (PS) externalization that eventually results in caspase activation [42]. In healthy cells, PS is confined to the plasma membrane’s inner leaflet. It is well known that when apoptosis is initiated, cells of all origins externalize PS on their outer side of the plasma membrane as a signal for phagocytosis [43]. Our results showed a significant amount of PS externalization after 24 h of cellular exposure to P3C (*p* < 0.05; Figure 3A), suggesting apoptosis as its mechanism of cell death.

Apoptosis is typically executed following activation of the intrinsic (mitochondrial) and/or extrinsic (death receptor) pathway depending on the stimulus. The intrinsic pathway is carried out on the mitochondria, eventually leading to a loss of membrane potential [42]. The extrinsic pathway is activated at the plasma membrane level upon ligation of a death receptor, triggering activation of caspase-8. Thus, to identify the apoptotic pathways induced by P3C, mitochondrial depolarization and caspase-8 activation were assessed. Our results indicated that P3C induced both mitochondrial depolarization and caspase-8 activation in TNBC MDA-MB-231 cells in a dose-dependent manner (Figure 3C and Figure 4B). These results support the notion that that P3C can activate both intrinsic and extrinsic apoptotic pathways.

The central organelle for ROS production in mammalian cells is mitochondria, and excessive ROS production elicits mitochondrial membrane depolarization [43]. Exposure to P3C caused a significant increase in ROS in MDA-MB-231 cells (Figure 4A) that potentially contributed to depolarizing the mitochondrial membrane potential to induce cell death. To corroborate that P3C triggered mitochondrial depolarization, caspase-3/7 activation was analyzed as a potential downstream apoptosis effector. Caspase-3/7 are enzymatic effectors required for the efficient execution of apoptosis [44]. Our data showed significant caspase-3/7 activation (*p* < 0.050) in P3C-treated MDA-MB-231 cells (Figure 3B). After caspase-3/7 activation, an important downstream caspase-3/7 substrate is poly- (ADP-ribose) polymerase (PARP), involved in DNA repair in response to cellular stress [45]. The PARP protein is an abundant nuclear enzyme (116 kDa) that, after being cleaved by caspase-3/7, produces a small (24-kDa) and a larger (89 kDa) fragment, which is considered a hallmark of apoptosis [46]. We investigated whether P3C could elicit PARP cleavage using a specific antibody (α-cleaved PARP-Alexa 488) and confocal microscopy. These immunocytochemistry studies indicate that P3C induces PARP cleavage in MDA-MB-231 cells, as evidenced by the presence of abundant nuclear green fluorescence signal (Appendix A). The aforementioned findings support and corroborate the conclusion that P3C induces the intrinsic apoptotic pathway.

Many anticancer drugs inflict their antiproliferative effects by causing cell cycle arrest, leading to apoptosis [47,48]. In the present study, we examined the potential cell cycle disruptive properties of P3C in MDA-MB-231 cells. Interestingly, P3C induced significant arrest in the G2/M and S phases (Figure 5C,D), accompanied by a noticeable reduction in the G0-G1 cycle with an increase in the sub-G0-G1 phase (Figure 5A,B), suggesting that the cytotoxic and apoptotic properties of P3C are due, in part, to a cell cycle arrest and DNA fragmentation.

Cell migration is crucial in multiple normal cellular physiological processes and many diseases, including the regeneration of injured tissues and cancer metastasis, respectively [49]. Furthermore, cell migration drives cancer metastasis, which is the major cause of high mortality of cancer patients (~90%) [50]. The scratch wounding assay is the most commonly used method to monitor cell migration/invasion in vitro and was used to determine whether P3C affects cell migration [51]. Using this assay, we evaluated the ability of P3C to interfere with the migration of MDA-MB-231 cells. Cell migration is usually assessed by comparing live-cell images captured immediately after the creation of a cell-free zone (time 0) with user-defined time intervals. Typically, the scratch-wound assay is based on brightfield illumination images. However, in this report, we included nuclear stain imaging (Hoechst 33342) combined with brightfield images to improve cell counting accuracy. Due to its permeability, the Hoechst blue-fluorescent DNA intercalator dye is suitable for staining living and dead cells. Nuclear staining provided higher precision in counting cell numbers, mainly when cell clusters were formed. Our results indicated that P3C effectively inhibited the metastatic migration of TNBC MDA-MB-231 cells in culture and was surprisingly more potent than doxorubicin (Appendix A).

The cytoskeleton is responsible for cell shape, migration, contraction, organization of the cytoplasm, cell polarity, and the formation and function of the mitotic spindle during cell division [52]. To further explore the mode of action of P3C, we chose to perform a microscopic analysis of the cytoskeleton on cells exposed to P3C for 2 h. The two major proteins forming the cytoskeleton, actin (microfilaments) and tubulin (microtubules), were detected and analyzed. Interestingly, our assays revealed a complete breakdown of the microtubules with the formation of aggregates (dots) in MDA-MB-231 cells (Figure 6E) and little to no damage in the actin fibers. When compared to paclitaxel, which is a well-known microtubule disrupting agent, P3C elicited more microtubule disruption. To extend this analysis, we evaluated cells undergoing mitosis, predicting an aberrant mitotic spindle. To facilitate the examination of cells experiencing mitosis, we used HeLa cells for these assays. Our results confirmed the phenotype previously seen in the MDA-MB-231 cells of disrupted microtubules displaying a dotted pattern in HeLa cells that were not undergoing cell division (Appendix A). Moreover, our results revealed that P3C-treated HeLa cells undergoing mitosis have a completely distorted mitotic spindle. In effect, no spindle fibers were observed, and instead, a dotted pattern was detected in all the dividing cells that were evaluated (Appendix A). It was easy to recognize HeLa cells undergoing mitosis due to DAPI staining, which binds to chromosomal DNA. In addition, phalloidin conjugated to Alexa-568 helped identify a typical actin organization in mitosis, the actomyosin cortex, where the microfilaments (F actin) form aggregates with myosin in the cell periphery. Thus, P3C caused chromosome mis-segregation and cytokinesis failure by interfering with mitotic spindle function, consequently inhibiting the cell cycle progression and cell migration. Overall, these data indicate that P3C acts as a microtubule-disturbing agent.

To better elucidate the molecular mechanism of P3C-mediated cytotoxicity, we performed a series of validated Luminex magnetic-bead-based Multiplex phosphoproteomic experiments. With this approach, we investigated the phosphorylation or dephosphorylation levels of numerous kinases potentially implicated in several biochemical and apoptosis-associated pathways. The Mitogen-Activated Protein Kinases (MAPKs) are serine/threonine protein kinases that participate in signal transduction pathways critical for cell metabolism, migration, pro-inflammatory mediators, differentiation, proliferation, and survival [53]. MAPKs are frequently elevated in several cancers. In this study, the phosphorylation levels of p38, ERK1/2, and JNK belonging to the MAPK family were analyzed in P3C-treated MDA-MB-231 cells. Our results showed that P3C treatment of cells decreased phosphorylation of both p38 and ERK1/2, whereas an increase in JNK phosphorylation was also noticed (Figure 7A–C). In general, p38/MAPK and JNK are activated to respond to extracellular genotoxic stressors to regulate cell survival by modulating cell cycle progression, inflammation, proliferation, and apoptosis [54,55]. Due to their roles in the aforementioned cellular processes, p38/MAPK and JNK-altered activity have vast tumor development effects. For instance, p38 is found up-regulated in several cancers, including breast, and is associated with metastasis [56]. Moreover, similar to our findings, crosstalk between p38 and JNK MAPK pathways has been reported. It has shown that chemical inhibition of p38 results in a large increase in JNK activation in epithelial cells and macrophages [57], and more recently in mouse models [58], which also appears to be the case for P3C in this study. The ERK cascades are responsible for many distinct cellular events, and their elevated expression occurs in most types of cancers [59]. Importantly, it has been described that constant ERK/MAPK pathway activation has anti-apoptotic effects that are induced by cell cycle inhibition promoting normal cell transformation into tumor cells [59,60,61]. Hence, ERK/MAPK pathway’s inhibition can prevent proliferation and induce apoptosis in tumor cells [59,62]. In this study, the decreased activity seen on both p38 and ERK1/2 can explain the P3C-mediated apoptosis induction, which could have been influenced by the observed disruption of cell cycle progression.

It is well known that MAPK signaling pathways control gene expression through the regulation of transcription factors. In this work, we confirmed that P3C alters the activity of several transcription factors that are likely induced by the upstream-modified activity of p38, ERK1/2, and JNK protein kinases. For these reasons, the phosphoproteomic profiles of CREB, STAT3, and NF-kB transcription factors were analyzed in P3C-treated MDA-MB-231 cells. Interestingly, our results showed a reduction in CREB, STAT3, and an increase in NF-kB phosphorylation (Figure 7D–F). The cyclic AMP (cAMP) response element-binding protein (CREB) is a transcription factor that participates in cell proliferation and differentiation [63]. CREB is overexpressed in most acute myeloid leukemia (AML) patients, supporting cell proliferation associated with an unfavorable prognosis [64]. Interestingly, polydatin, a compound isolated from *Polygonum cuspidatum* (knotweed), inhibits CREB activity by decreasing its phosphorylation and reducing the proliferation of human breast cancer cells [65]. Furthermore, Signal Transducers and Activators of Transcription (STATs) comprise a family of seven transcription factors that, upon activation, influence a variety of cellular responses, including apoptosis [66]. STAT3 is commonly overexpressed and constitutively activated in numerous malignancies, promoting cancer, inducing cell proliferation, tumor-mediated immune evasion, angiogenesis, and cell survival [67]. STAT3 activation is most prominent in breast cancer tissues and is found to be elevated in greater than 40% of breast cancer cases [68], making it a promising molecular target in the development of new TNBC therapies. In this study, the decreased phosphorylation of CREB and STAT3 can reduce cell proliferation and consequently induce cell death in P3C-treated MDA-MB-231 cells.

The Nuclear Factor kappa-light-chain-enhancer of activated B cells (NF-kB) is an inducible transcription factor that functions in inflammatory responses, cellular growth, apoptosis, proliferation, and survival [69]. Increased NF-kB activity elicits expression of target genes able to lead to apoptotic resistance and has been linked to tumor resistance in anticancer therapy [70]. In contrast, inhibition of NF-kB phosphorylation resulted in an increased sensitivity of cancer cells to the apoptotic activities of chemotherapeutic agents [71]. The activation of CREB, STAT3, and NF-kB supports cell proliferation, differentiation, and survival. However, our data indicate a reduction in CREB, STAT3 and an increase in NF-kB phosphorylation induced by P3C. Thus, it appears that CREB and STAT3 signaling pathways are more critical for the induction of cell death in MDA-MB-231 cells than NF-kB. Accordingly, our results revealed a significant down-regulation in phosphorylation of p38, ERK1/2, CREB, STAT3 signaling pathways during the P3C-induced apoptosis. Among the detected activated pathways, both JNK and NF-kB protein phosphorylation levels were insufficient to rescue the cytotoxicity caused by P3C. It is also possible that P3C-mediated JNK-phosphorylation resulted in a negative survival effect by eliciting inhibition of STAT3 phosphorylation, as previously described [72]. 

To gain additional insights into the potential mechanisms affected by P3C treatment, Akt and P70S6K protein kinases were also studied. The phosphorylation of the Akt serine/threonine-protein kinase is essential for the transduction of the Lyn/PI3K/Akt signaling pathways, which serves as a strong indicator implicated in pro-survival, cell signaling, and cytoprotection [19,73]. In addition, P70 ribosomal S6 kinase (P70S6K) is a serine/threonine kinase acting as a downstream effector of the mammalian target of rapamycin (mTOR) signaling pathway involved in cell proliferation, cell growth, cell-cycle progression, and glucose homeostasis [74]. Both essential kinases, Akt and P70S6K exhibited a reduction in phosphorylation (66% and 51% reduction at 10 µM, respectively) in P3C-treated cells. However, these effects were not statistically significant due to prominent standard deviations. Nonetheless, the observed dephosphorylations can contribute with additive or synergistic effect to the P3C cytotoxic mechanism of action (Appendix A).

The Src family is a group of cytoplasmic non-receptor tyrosine kinases that play a critical role in many important biological processes, including proliferation and cell death [75]. For this reason, we explored the capacity of P3C to alter the phosphorylation levels of two Src family tyrosine kinases, Fyn and Src, in MDA-MB-231 cells. Fyn is a membrane-associated tyrosine kinase, and its activation has been linked to cell growth progression [76]. In addition, Fyn activation leads to keratinocyte growth arrest and inhibition of differentiation, involving cell signaling and programmed cell death in vivo [77,78]. Our results indicated that P3C induced Fyn dephosphorylation (~60% at 1 µM), which could be associated with increased cell death in MDA-MB-231 cells (Figure 8A). Interestingly, P3C did not significantly affect the phosphorylation of Src, suggesting it does not play a role in this process.

As described in the previous sections, P3C decreased phosphorylation levels of p38, ERK1/2, CREB, STAT3, Akt, P70S6K, and Fyn, which collectively could lead to the gene expression profiles seen in our transcriptome analyses that revealed hundreds of genes that were affected by treatment with P3C.

mRNA sequencing (RNA-Seq) is an accurate approach to transcriptome profiling that utilizes the power of next-generation sequencing technologies. In this study, the transcriptome data obtained from two P3C-treated TNBC cell lines, MDA-MB-231, and MDA-MB-468, showed different gene expression profiles. As can be seen in Figure 8 and Figure 9A,B, P3C treatment resulted in 177 and 478 up-regulated and 249 and 89 down-regulated DEGs, in MDA-MB-231 and MDA-MB-468, respectively. However, a closer look into these profiles revealed 28 DEGs (23 up- and five down-regulated) that were commonly affected (up- and down-regulated) in the two cell lines tested (Figure 10). Gene ontology and ingenuity pathway analyses were performed on the 28 genes in the attempt to find common mechanisms of P3C cell death induction in the two cell lines analyzed. In both P3C-exposed MDA-MB-231 and MDA-MB-468 cell lines, several genes that were similarly altered are implicated in biological processes and signaling pathways that coincide with the in vitro experimental findings in this study. These altered genes were implicated in biological processes such as apoptosis induction, response to stress, regulation of phosphorylation, and inactivation of MAP kinase activity (Appendix A; highlighted terms). Moreover, the SAPK/JUNK, STAT3, ERK/MAP, P38 MAPK signaling pathways, and cell cycle regulation were relevant canonical pathways associated with our in vitro experimental findings (Table 3).

In an attempt to correlate the transcriptome (RNA-seq) profiles with our in vitro experimental findings, we observed that several genes/pathways match well to the effects detected after P3C treatment. For instance, B-cell Translocating Gene 2 (*BTG2*) gene overexpression seen via RNA-seq in the two cell lines studied (P3C-treated; Figure 10) reinforced the antiproliferative observation obtained from the arrest in S and G2/M phases during the cell cycle experiments (Figure 5). *BTG2* has been implicated as a tumor suppressor gene as it is frequently deleted or mutated in cancer malignancies [79]. The *BTG2* expression induced G2/M arrest and has also been implicated in anti-metastasis, anti-invasion, and mitochondrial depolarization [79,80]. In addition, *BTG2* up-regulation supports the anti-metastatic and anti-invasive results obtained during the scratch-wound assay for cell migration (Appendix A; *p*-value of 0.00001) and also confirms its participation in the mitochondrial depolarization detected through the flow cytometry assay (Figure 4B).

The transcriptome analyses also detected the up-regulation of *DUSP4* and *DUSP10* genes in the two P3C-treated TNCB cell lines tested (Figure 10C). The DUSP4 and DUSP10 proteins are members of the dual-specificity protein phosphatase subfamily [81,82] that regulate the MAP kinase superfamily. DUSP4 preferentially inactivates MAPK/ERK1/2, SAPK/JNK, and p38, whereas DUSP10 dephosphorylates p38 and SAPK/JNK [81]. *DUSP4* and *DUSP10* overexpression suggest a direct correlation with the inactivation of ERK1/2 and p38 kinases seen during our Multiplex immunoassays (Figure 7), leading to cell growth arrest and cell death [83,84]. Similarly, the proto-oncogene *PIM1* gene was up-regulated, which encodes a serine/threonine kinase that plays a crucial role in controlling cell proliferation, apoptosis, and migration. *PIM1* also participates in the JAK/STAT pathway, which is vital in survival pathways, including STAT3 and NF-kB transcriptional complexes [85,86]. The overexpression of *PIM1* seen in this study could be involved in the upregulation of NF-kB (Figure 7E). However, it does not appear to have the same effect on STAT3 expression as this protein was down-regulated after P3C treatment. However, *PIM1* overexpression could be implicated in the P3C-induced antiproliferative and apoptosis-inducing effects (Table 3).

Transcriptome analysis revealed a P3C-induced downregulation of the *PIF1* helicase gene that maintains nuclear and mitochondrial DNA integrity in eukaryotes [87]. *PIF1* downregulation could be associated with the mitochondrial damage leading to its depolarization and the nuclear DNA fragmentation detected during the cell cycle in vitro analyses (Figure 4B and Figure 5). Additionally, the Tumor Necrosis Factor Receptor Superfamily Member 10D (*TNFRSF10D; TRAIL-R4;* TNF-receptor superfamily) gene was found up-regulated (RNA-seq) upon P3C treatment. Unlike other members of the TNF family, it contains a truncated cytoplasmic death receptor domain and does not induce apoptosis but may play an inhibitory role in TRAIL-induced cell death. This receptor has been shown to induce NF-kB and AKT activation and has pro-tumoral functions [88,89]. Although this receptor could be responsible for NF-kB activation (Figure 7F), it does not appear to interfere with the promotion of apoptosis.

N-Myc downstream-regulated gene (*NDRG1)* was also found to be up-regulated after P3C treatment, and this protein is known to be necessary for p53-mediated caspase activation and apoptosis [90]. Moreover, *NDRG1*, an iron-regulated growth and metastasis suppressor, was negatively correlated with cancer progression in several tumors [91]. *NDRG1* has noticeable anti-oncogenic activity associated with reduced cell proliferation, migration, invasion, and angiogenesis [91]. *NDRG1* up-regulation correlates with caspase activation, reduced migration, and antiproliferative effects of P3C in our in vitro assays. Additionally, the *NDRG1* gene has also been shown to protect cells from spindle disruption, regulating the microtubule dynamics and maintaining euploidy [39]. The up-regulation of *NDRG1* observed in this study could have been induced to diminish the microtubule damage caused by P3C treatment.

Lastly, the proline- and serine-rich coiled-coil 1 (*PSRC1*) gene, also known as *DDA3*, is required for normal mitosis progression during anaphase [38]. *PSRC1* regulates mitotic spindle dynamics, increasing the turnover rate of microtubules on metaphase spindles [37]. Remarkably, this *PSRC1* gene was found down-regulated on both cell lines tested (Figure 10C), suggesting its importance in the observed disruption of microtubule structure after P3C treatment (Figure 6E, Appendix A). Down-regulation of this gene could have also contributed to the arrest in the G2-M phase during the cell cycle analysis (Figure 5), where an intact microtubule spindle formation is required for the normal progression of mitosis.

Our transcriptome results generally suggest a deeper insight into understanding the molecular mechanism responsible for P3C cytotoxicity inflicted in TNBC cells. Additionally, our findings indicate that P3C perturbs an intricate series of biochemical pathways to inflict its cytotoxicity, implicating the ERK1/2-MAPK, SAPK-JNK, p38-MAPK, JAK-STAT, PIF1 DNA repair, BTG2, PSRC1 mitosis, and NF-kB pathways, among others (Table 3). 

## 5. Conclusions

In conclusion, results from this study indicate that P3C has great potential as an anticancer drug for the treatment of TNBC; further studies should be performed to evaluate its effectiveness in-vivo. P3C demonstrated potent cytotoxicity against 26 human cancer cell lines of distinct tissue origins. It triggered both the intrinsic and extrinsic apoptotic pathways in TNBC cells and induced oxidative stress leading to mitochondrial depolarization and caspase 3/7 and -8 activation. It also altered cell cycle progression by arresting cells at the S and G2/M phases inducing PARP cleavage and DNA fragmentation. This last effect can be attributed to the P3C-induced microtubule disruption seen in this study, which revealed a disorganized mitotic spindle that blocked cytokinesis. In addition, it was observed that P3C modified kinase activity through the dephosphorylation of several kinases, including p38, ERK1/2, CREB, STAT3, and Fyn. In agreement with these findings, the gene expression profiles induced by P3C on two TNBC cell lines verified a correlation with apoptosis, oxidative stress induction, kinase activity inhibition, and microtubule array disruption. Thus, P3C primarily inhibits mitosis and multiple kinase activities to induce apoptosis. The discovery of P3C provides a foundation for developing similar highly active anticancer drugs with activity against TNBC and other types of cancer cells.

## Figures and Tables

**Figure 1 cells-11-00254-f001:**
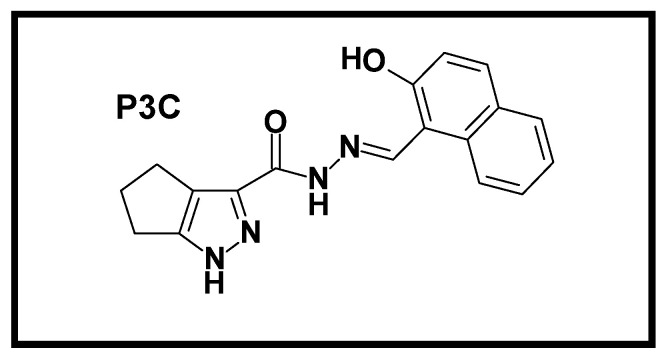
Chemical structure of P3C, N’-[(2-hydroxy-1-naphthyl)methylene]-1,4,5,6-tetrahydrocyclopenta[c]pyrazole-3-carbohydrazide.

**Figure 2 cells-11-00254-f002:**
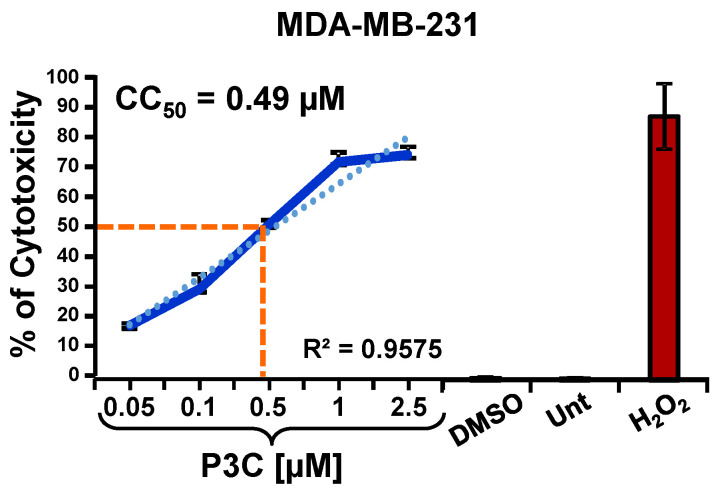
Dose-response curve (continuous blue lane) and CC_50_ value of P3C determined via the Differential Nuclear Staining (DNS) assay on MDA-MB-231 cells. Cells were exposed for 72 h to a concentration gradient of P3C, as indicated on the *x*-axis, whereas in the *y*-axis, it shows the cytotoxicity percentage (dead cells). In this experiment, several controls are depicted (bar graphs): as a negative control, untreated (Unt) cells; the diluent of experimental compounds, DMSO, as contained in the experimental samples (0.5% *v*/*v*); and as a positive control of cytotoxicity, H_2_O_2_ (1 mM). Cell viability was examined in a live-cell manner via IN Cell Analyzer 2000 system and IN Cell Analyzer Workstation 3.2 software (GE Healthcare). Each experimental point represents the average of five independent measurements and error bars their corresponding standard deviation. Cytotoxic concentration 50% (CC_50_) in micromolar (µM) units is defined as the concentration of chemical compound required to disrupt the plasma membrane of 50% of the cell population after 72 h of incubation, as detailed in Materials and Methods. The CC_50_ was achieved by using a linear interpolation calculator software and is denoted at the point of convergence of the orange dashed lines in the figure (https://www.johndcook.com/interpolator.html, accessed on 10 January 2022). The dotted blue line indicates the R^2^ linear regression analysis, 0.9575.

**Figure 3 cells-11-00254-f003:**
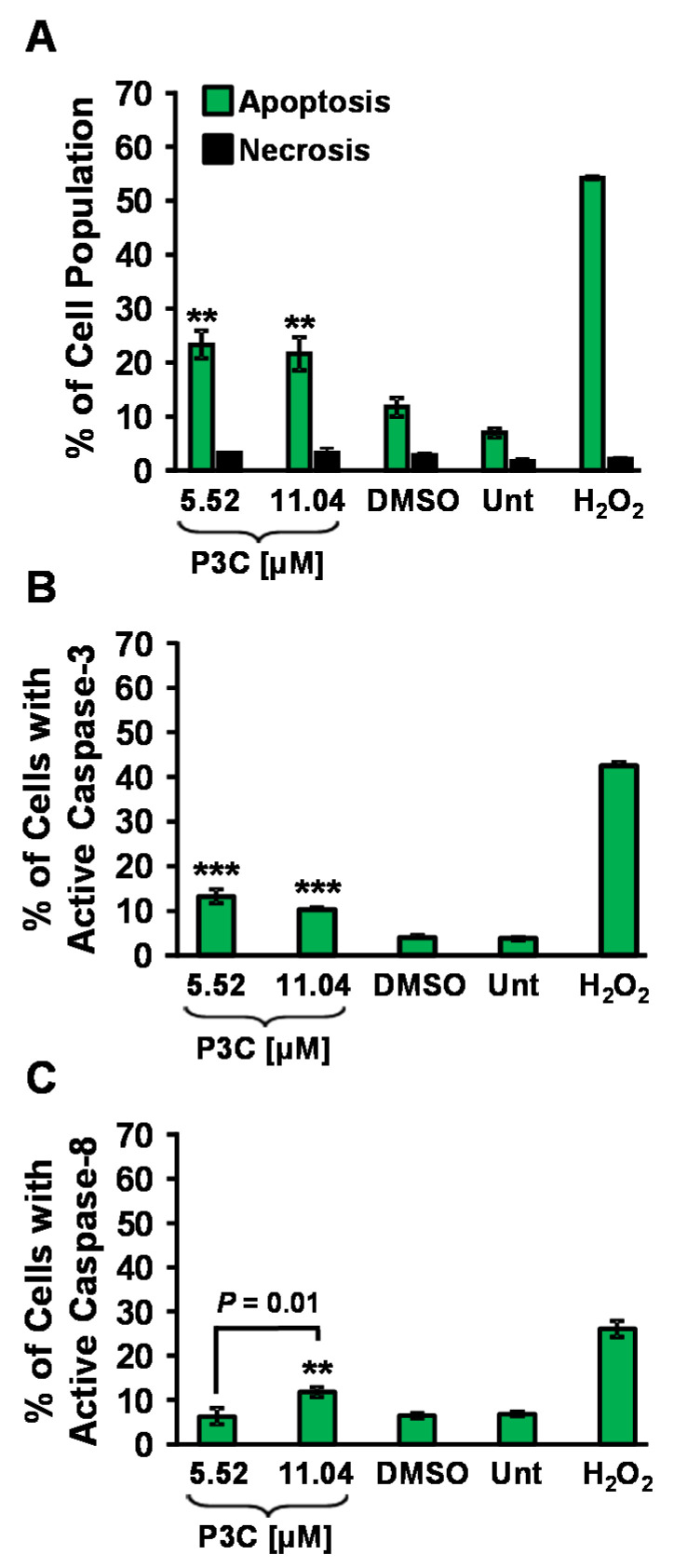
P3C cell death induction is mediated through apoptosis via intrinsic and extrinsic pathways. (**A**) Evaluation of phosphatidylserine externalization through flow cytometry indicated that P3C induced apoptosis in MDA-MB-231 cells. Analysis was performed after 24 h of exposure to P3C. Percentages shown represent the total apoptosis values (early and late apoptosis, Appendix A); (**B**,**C**) P3C-treated MDA-MB-231 cells exhibited a significant amount of caspase-8 activation after 4 h of incubation (Appendix A), whereas caspase-3/7 activation was detected after 8 h of incubation (Appendix A). Cells with a positive fluorescence for NucView 488 caspase-3 enzyme substrate denoted caspase-3 activation, whereas, cells indicating positive fluorescent levels of FITC-IETD-FMK marker designated caspase-8 activation, both measurements analyzed through flow cytometry. For these experiments (**A**–**C**), cells were treated with P3C CC_50_ (5.52 µM) and 2× CC_50_ (11.04 µM), also 1% DMSO, 1 mM H_2_O_2_ as vehicle and positive control for cytotoxicity, respectively, as well as untreated controls. Averages of three technical replicates are shown in percentages, and standard deviations are denoted on each bar. Statistical evaluation on data was accomplished by using two-tailed Student’s paired *t*-test, and the asterisk annotations in each graph represent statistical significance of the treatments against the vehicle control (**) *p* < 0.01 and (***) *p* < 0.001.

**Figure 4 cells-11-00254-f004:**
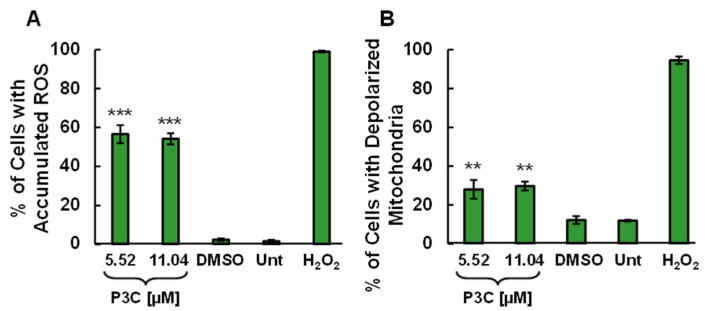
P3C induces ROS accumulation and loss of mitochondrial membrane potential in MDA-MB-231 cells. (**A**) A significant amount of ROS accumulation was observed in MDA-MB-231 cells after 18 h of treatment with P3C (Appendix A). Treated cells were stained with 10 mM of carboxy-H_2_DCFDA fluorescein reagent to quantify ROS by flow cytometry; (**B**) Mitochondrial membrane depolarization was identified in MDA-MB-231 cells after 8 h of exposure to P3C (Appendix A). In these experiments (**A**,**B**), cells were treated with P3C CC_50_ (5.52 µM) and 2× CC_50_ (11.04 µM). In addition, a vehicle (DMSO), untreated (Unt), and H_2_O_2_ positive controls were included. Percentages represent averages of three technical replicates, and standard deviations are shown on each bar. Two-tailed Student’s paired *t*-test was employed for statistical analysis, and the asterisk annotations in each graph represent statistical significance of the treatments against the vehicle control; (**) *p* < 0.01 and (***) *p* < 0.001.

**Figure 5 cells-11-00254-f005:**
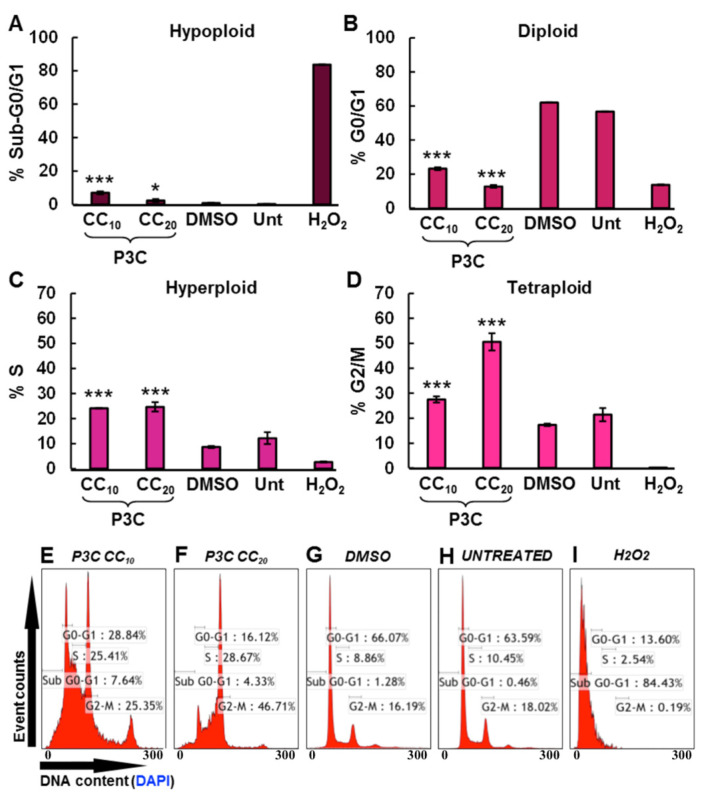
P3C disrupts cell cycle progression by arresting MDA-MB-231 cells in S and G2-M phases. P3C alters cell cycle progression in MDA-MB2-31 cells after 72 of exposure to P3C CC_10_ (1.1 µM) and CC_20_ (2.2 µM). DNA fragmentation and cell cycle arrest was observed in S and G2-M phases, denoted by a significant increase in P3C-treated cells in the (**A**) sub G0-G1, (**C**) S and (**D**) G2-M phases. And, as a consequence the P3C- treated cells denoted a reduction in the (**B**) G0-G1 phase. After treatment, cells were stained with NIM-DAPI and subsequently analyzed by flow cytometry. The FL9 detector was used to identify DAPI bound to DNA that allows the quantification of DNA content. 1% DMSO and 1 mM of H_2_O_2_ were used as a vehicle and positive control for death, respectively. Statistical significance of the experimental treatments with the vehicle control (DMSO) are denoted with the following symbols: (*) *p* < 0.05; and (***) *p* < 0.001. Bars represent an average of three independent measurements, and standard deviations are indicating the experimental variability; (**E**–**I**) Representative flow cytometric dot plots used to quantify the percentages of each cell cycle phase. Approximately 10,000 events (cells) were acquired and analyzed per sample using a Gallios flow cytometer and Kaluza 1.3 software (Beckman Coulter).

**Figure 6 cells-11-00254-f006:**
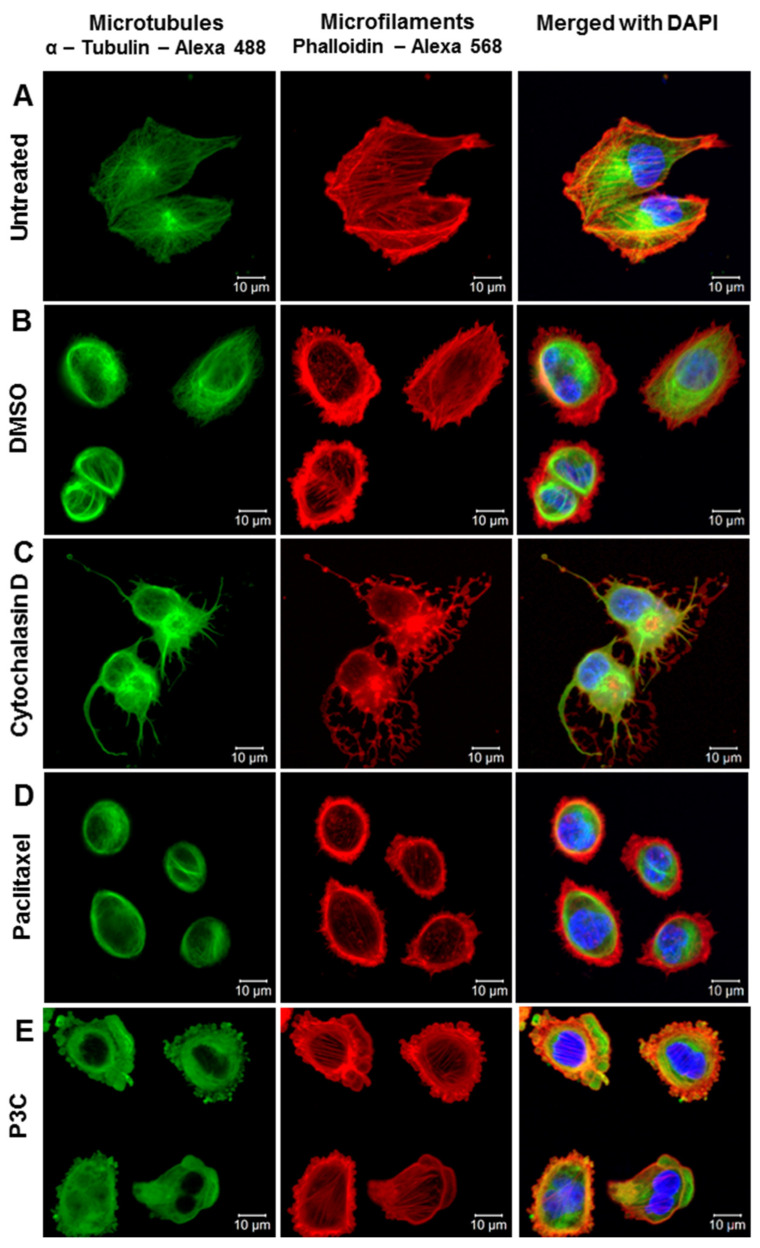
P3C perturbs tubulin organization on MDA-MB-231 cells after 2 h of exposure. Untreated (**A**) and treated (**B**–**E**) cells were stained with α-tubulin-Alexa-488 (microtubules), phalloidin-Alexa-568 (microfilaments), and DAPI (nucleus), and confocal immunofluorescent analysis was performed. P3C CC_50_ (**E**) was used for this experiment along with paclitaxel (1 µM) (**D**), and cytochalasin D (5 µg/mL) (**C**), which were included as microtubule and microfilament disruptive agents, respectively.

**Figure 7 cells-11-00254-f007:**
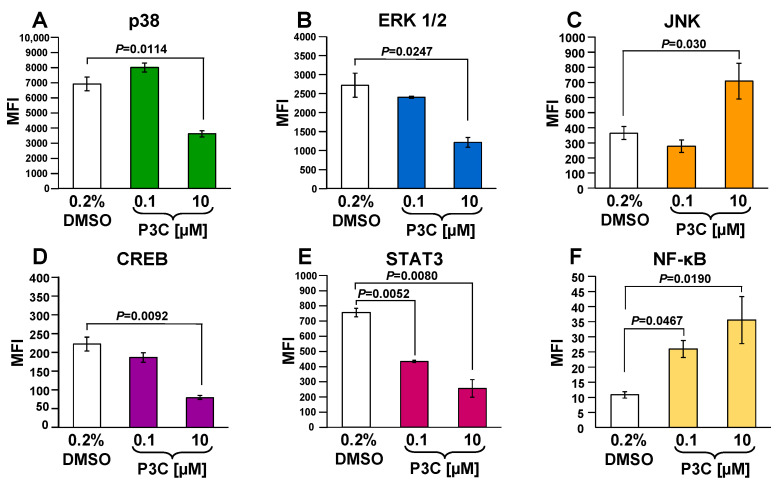
P3C decreased CREB, p38, ERK, and STAT3 and increased JNK and NF-kB phosphorylation in MDA-MB-231 cells after 3 h of exposure. Cellular extracts from cells exposed with 0.1 and 10 µM of P3C were analyzed with the antibody/bead-based Luminex xMAP technology. Control cells were exposed to solvent (0.2% *v*/*v* of DMSO) under the same conditions and analyzed concomitantly. The *y*-axis shows the median fluorescence intensity (MFI), and each error bar represents standard deviations of the mean of two independent measurements. Statistical significance was determined using Student’s *t*-test with values of *p* ≤ 0.05 deemed significant. Data acquirement and analysis were accomplished via xPONENT 3.1 software (Luminex).

**Figure 8 cells-11-00254-f008:**
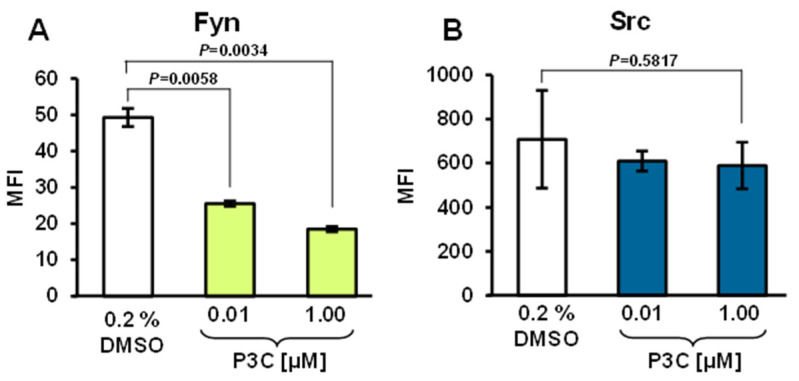
P3C treatment decreased Fyn (**A**) but not Src phosphorylation (**B**) in MDA-MB-231 cells after 3 h of exposure. Intracellular Bead-Based Multiplex Assay via the Luminex technology was employed for analysis of cell lysates that were previously exposed to 0.01 and 1 µM of P3C for 3 h. 0.2% *v*/*v* of DMSO was included as solvent control. The median fluorescence intensity (MFI) is shown in the *y*-axis. Each bar represents the average of two independent measurements, each with two technical replicates, and error bars the standard deviations (*n* = 4). Statistical significance was determined using Student’s *t*-test. A value of *p* ≤ 0.05 was deemed significant. Data acquirement and analysis were accomplished via xPONENT 3.1 software (Luminex).

**Figure 9 cells-11-00254-f009:**
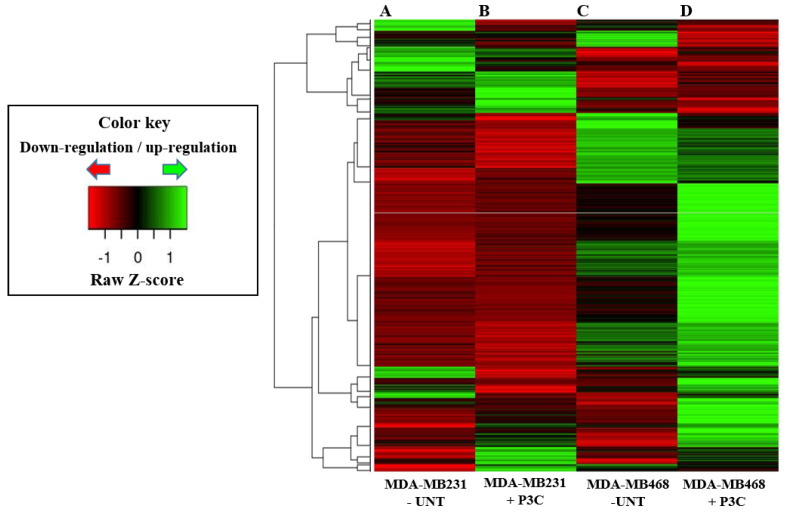
Gene expression profiles of (**A**,**B**) MDA-MB-231 and (**C**,**D**) MDA-MB-468 cell lines treated with or without P3C. The heat map shows the expression values at a one-time point (6 h). Treatments and/or untreated samples were performed in triplicate. Each row represents data from one transcript, and each column corresponds to untreated or treated samples for each cell line tested. The color legend of the row-wise normalized expression values (Z-scores) is given on the left.

**Figure 10 cells-11-00254-f010:**
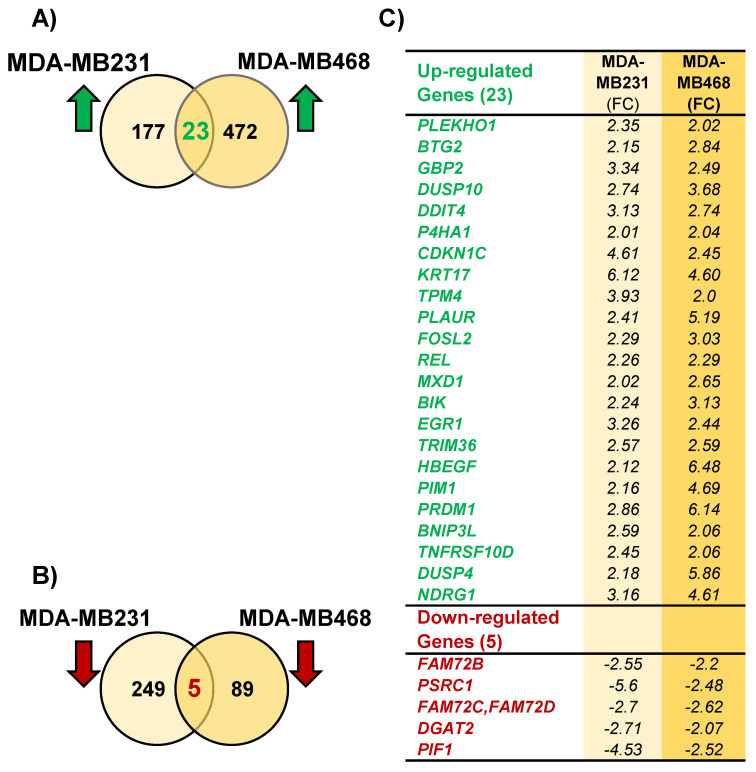
Comparison of transcriptome sequencing (RNA-Seq) data of two P3C-treated TNBC cell lines; MDA-MB-231 and MDA-MB-468. Venn diagrams depict the numbers of common (overlapping circles) and uniquely differentially expressed transcripts (non-overlapping circles) from two TNBC cell lines (**A**,**B**). Venn diagrams display 23 up-regulated (green) (**A**) and 5 down-regulated (red) (**B**) genes commonly altered between the two cell lines tested. The list of shared up-, and down-regulated genes are displayed in panel (**C**), and fold change (FC) values for each gene are shown. Genes selected for this analysis were those differentially regulated by either >2.0 (overexpressed), or <−2.0 (down-regulated) fold, as compared to the solvent (30%/70% *v*/*v* PEG-400/PBS) treated cells.

**Table 1 cells-11-00254-t001:** CC_50_ and SCI values of P3C in a panel of 27 different cancer and one non-cancerous cell line at 48 h (bottom) and 72 h (top) of exposure.

Cell Line	Tissue Origin	CC_50_ (µM) *	SCI **	*p*-Value ^‡^
		**72 h**		
Hs27	Normal foreskin epithelial	3.47 +/− 0.45	1	n/a
MDA-MB-231	Breast adenocarcinoma	0.49 +/− 0.12	7.1	0.0003
MDA-MB-468		0.25 +/− 0.02	13.9	0.0002
HCC70	Breast carcinoma	5.40 +/− 0.084	0.6	0.001
HCC1419		6.27 +/− 0.14	0.5	0.0005
MCF-7		0.45 +/− 0.014	7.7	0.0003
T47D		0.78 +/− 0.07	4.5	0.0005
OVCAR-8	Ovarian carcinoma	0.44 +/− 0.07	7.9	0.0003
OVCAR-5		0.60 +/− 0.008	5.8	0.0003
OVCAR-3	Ovarian adenocarcinoma	0.30 +/− 0.03	11.6	0.0002
OV-90	Metastatic ovarian adenocarcinoma	0.65 +/− 0.19	5.3	0.0005
NCI-H358	Non-small cell lung cancer	0.19 +/− 0.11	18.3	0.000254
NCI-H460	Large cell lung cancer	0.67 +/− 0.17	5.2	0.000545
A549	Lung carcinoma	0.27 +/− 0.03	12.9	0.000262
PC-3	Prostatic adenocarcinoma	1.41 +/− 0.09	2.5	0.00147
LnCap	Prostatic carcinoma	0.79 +/− 0.09	4.5	0.000538
PANC-1	Pancreatic carcinoma	0.76 +/− 0.10	4.6	0.000524
A375	Melanoma	0.40 +/− 0.03	8.7	0.000296
WM-115		0.38 +/− 0.05	41.3	0.000292
		**48 h**		
Hs27	Normal foreskin epithelial	5.76 +/− 0.36	1	n/a
CEM	T lymphoblastic leukemia	0.48 +/− 0.02	12	0.000014
MOLT-3	Acute lymphoblastic leukemia	6.54 +/− 0.36	0.9	0.05676
HL-60	Acute promyelocytic leukemia	0.58 +/− 0.05	9.9	0.000016
JURKAT	Acute T cell leukemia	0.37 +/− 0.01	15.6	0.000013
NALM-6	Lymphoblastic leukemia	4.91 +/− 0.33	1.2	0.03937
RAMOS	Burkitt’s lymphoma	0.31 +/− 0.009	18.6	0.000013
RPMI-8226	Myeloma	4.45 +/− 0.091	1.3	0.003631
MM.1S		0.96 +/− 0.014	6	0.000021
MM.1R	Multiple Myeloma	0.76 +/− 0.01	7.6	0.000018

* The CC_50_ refers to the cytotoxic concentration at which 50% of the cell population dies, either after 48 or 72 h of drug exposure. +/− values denote standard deviations. ** Selective Cytotoxicity Index (SCI) was determined by dividing the CC_50_ of Hs27 by the CC_50_ of the individual cancer cell lines. ^‡^
*p*-values were calculated using non-cancerous normal fibroblast Hs27 as reference vs. each individual cancer cell line tested. n/a = non applicable.

**Table 2 cells-11-00254-t002:** Relevant canonical pathways associated with the 28 DEGs in common in MDA-MB-231 and MDA-MB-468 cells.

Ingenuity Canonical Pathways	Genes	*p*-Values
SAPK/JNK Signaling	*DUSP10,DUSP4*	6.46 × 10^−3^
Cell Cycle Regulation	*BTG2*	4.27 × 10^−2^
p38 MAPK Signaling	*DUSP10*	8.13 × 10^−2^
STAT3 Pathway	*PIM1*	1.31 × 10^−1^
ERK/MAPK Signaling	*DUSP4*	1.48 × 10^−1^

**Table 3 cells-11-00254-t003:** Correlation of some relevant P3C-altered genes (up- and down-regulated via RNA-seq), with their functions, our in vitro assays, and their biochemical pathways.

RNA-Seq	Gene Function	Supporting In Vitro Assay	Pathway
**↑** ** *BTG2* **	Cell cycle regulation and Antiproliferation	Arrest in S & G2/M phases during cell cycle by FC *	BTG2 Cell cycle and apoptosis
	Anti-metastasis and Anti-invasive	Scratch wound assay by CM ^‡^	
	Mitochondrial depolarization	JC-1 assay *via* FC	
**↑** ** *DUSP4* ** **↑** ** *DUSP10* **	Dephosphorylation of ERK1/2, SAPK/JNK, and p38 MAP Kinases	Inactivation of ERK1/2 and p38 during multiplex cell signaling assays	ERK1/2, SAPK/JNK and p38 MAPK
**↑** ** *PIM1* **	Cell proliferation	Cell cycle arrest found by FC	JAK/STAT
	Apoptosis	Apoptosis assay *via* FC	
	Migration	Scratch wound invasive assay by CM analysis	
**↓** ** *PIF1* **	Maintenance of nuclear and mitochondrial DNA	Cellular DNA fragmentation detected during cell cycle FC analysis	PIF1 DNA repair
**↑** ** *TNFRSF10D* **	Apoptosis	Apoptosis assay *via* FC	NF-κB
	Signals NF-κB activation	Activation of NF-κB found *via* Multiplex cell signaling assays	
**↑** ** *NDRG1* **	Caspase activation	Caspase 3/7 and 8 assays by FC	NDRG1 cell cycle, apoptosis
	Antiproliferation and Apoptosis	Cell cycle and apoptosis assays *via* FC	
	Anti-metastasis and Anti-invasive	Scratch wound assay by CM ^‡^	
**↓** ** *PSRC1* **	Microtubule maintenance and stability, including regulation of spindle during mitosis	Disturbance of microtubules and a deformed spindle unable to undergo mitosis found by immunohistochemistry and CM	PSRC1

* FC = Flow cytometry; CM ^‡^ = confocal microscopy; **↑** Gene up-regulation; **↓** Gene down-regulation.

## Data Availability

Not applicable.

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
