# Peer review of "Identification of a Potent Cytotoxic Pyrazole with Anti-Breast Cancer Activity That Alters Multiple Pathways"

_cells, 2022, doi:10.3390/cells11020254_

Round 1

Reviewer 1 Report

Comments to the Author:

The manuscript Nr. Cells-1468263 entitled: “Identification of a potent cytotoxic pyrazole with anti-breast cancer activity that alters 3 multiple pathways”, by Gutierrez et al., describes a novel pyrazole-based derivative (P3C) that displayed potent cytotoxicity, particularly in two triple-negative breast cancer (TNBC) cell lines. Using substantive and good developed in vitro assays exhaustive analysis was done.  The more substantial analysis is needed to be valid for final publication in the Cells (MDPI) Journal.

Major concerns:

  1. P3C was initially identified as a cytotoxic pyrazole from a high-throughput screen (HTS) using a human TNBC MDA-MB-231 cell line. The manuscript would benefit tremendously when some “normal” cells (fibroblasts or hepatocyte assays) would be involved to prove cytotoxicity of compound specifically on cancer cells and not on “normal” surrounding cells including a broad range of compound concentrations and especially from 0.25 to 0.49 µM range.

  1. Some standard functional assays (colony-formation assays or 7-day effect analysis) are appreciated in such new compound determination and definition. Can you include some longer time points of analysis including P3C in low and high concentration ranges?

  1. Kinase Multiplex Assays and transcriptome analysis were used for clarifying the mechanism of action of the P3C derivative. Unfortunately, the activity is not specific and rather broad on different cellular effects that make such derivatives not defined as specific and targeted compounds. It worries that MAPK activity is not specifically defined and maybe one of the problems is that compound used in IC50s range is too high and unspecific, therefore the analyses in lower concentration range with specific antibody activity detection of MAPK activity is preferred for manuscript final improvement.

Minor concerns:

  1. Line 15: CC50s should be defined in the abstract
  2. Rewording of lines 176-179. Sentence is confusing
  3. Line 205: the number of the indicated section should be mentioned
  4. Line 224-225: the sentence is not understood and probably not required for that section?
  5. Figure 2 legend (line 439-440) says MDA-MB-231 cells exhibited a significant amount of caspase -3/7 and -8 activation after 8 and 4 h of exposure to P3C, respectively. However, in the figure, it is not shown which results are after 4 and 8 h of exposure.
  6. In figure 7, it was mentioned in the legend that only 2 biological replicates were analyzed. For the statistics with error bars and significance definition, this is not sufficient for conclusive results. Experiments could be repeated at least one more time.
  7. Figures 1 to 7 show results of MDA-MB-231 cells treated with P3C while Figures 8&9 shows results for both MDA-MB-231 and MDA-MB-468. Why were both cell lines not initially tested for the cytotoxicity of P3C?
  8. The time-points after treatment with P3C are not consistent. Mechanism of cell death and PARP cleavage was 24 h after P3C exposure, ROS production after 18 h, cell cycle after 72 h, microtubule structure after 2 h, multiplex immune assays after 3 h, gene expression profile after 6 h. What is the rationale behind choosing the different time points? What is the real effect after long-time point analysis including 72 hours and 7 days?

Reviewer 2 Report

-The citation to Differential Nuclear Staining (DNS) assay (17) seems to be irrelevant. Please include explanations about the screening assay. 

- Dose-response graphs including the R2 values are required. A supplementary can be helpful.

- Abbreviations are required to be defined when first mentioned in the text.

- SCI has been defined twice. Please check for other redundancies.

- P3C has a very toxic effect on the normal cell line. This cannot be justified by using a second index, SCI. From the cytotoxicity perspective, the molecule does not have a selective killing effect on cancer cells. The CC50 values are all quite low. 

- Is there any significant differences between the reported CC50 values in different cell lines?

- Flow cytometric histograms and graphs are necessary to justify the cell death mechanism, ROS production, mitochondrial membrane potential, cell cycle analysis. Otherwise, the bar charts cannot be accepted.

- Since P3C is highly potent and has a very low CC50, the application of this molecule for migration/invasion assay is not justified. The drug is hghly cytotoxic, it kills the cells and of course the migration and invasion is inhibited.

- The exposure times in different experiments do not match. CC50 values are reported at 48-72 h and some of the experiments are porformed within 2 h of exposure.

Reviewer 3 Report

Researchers are still looking for compounds with promising anticancer activity. The study presents interesting findings and is worth publication in Cells. Authors put effort to check the cytotoxic properties of compound in 27 cell lines to choose one for further biological studies.

I have some minor comments to this paper.

  1. Authors should explain in details the reason for their choice of MDA-MB-231 cancer cell line. The most sensitive for new compound was MDA-MB-438 and NCI-H358 with the lowest CC50.  More detailed explanation should be included in the paper.
  2. What was the reason for choice of the doses (5.52 and 11.02), when CC50 was 0.49?
  3. The higher dose of compound should increase the apoptotic effect. Was a statistically significant difference between the doses of the tested agent? Could you explain it? (Fig. 2A and B, Fig.3)
  4.  The one experiment includes HeLa cells, other experiments don't include it. What was the reason for it? 
  5. The study is based on MDA-MB-231, but Fig.8 and Fig.9 present also results from MDA-MB-438. I think that authors should only focus on MDA-MB-231.
  6. After minor revision the paper will be suitable for publication.

Round 2

Reviewer 1 Report

The manuscript Nr. Cells-1468263 entitled: “Identification of a potent cytotoxic pyrazole with anti-breast cancer activity that alters 3 multiple pathways”, by Gutierrez et al., has improved after revision with complete response to the reviewer comments, and therefore I agree with its acceptance and publication in the Cells MDPI journal.

Reviewer 2 Report

The research design has systematic problems that have not been/cannot be addressed. 

Moreover, CC50 should be calculated as instructed by Graphpad Prism, using non-linear regression, the flow cytometery histograms are all included in the supplementary file, etc.